**Molecular composition of organic aerosols in central Amazonia: an ultra-high resolution mass spectrometry study**

I. Kourtchev[1]*, R.H.M. Godoi[2], S. Connors[1], J.G. Levine[3], A. Archibald[1,4], A.F.L. Godoi[2], S.L. Paralovo[2], C.G.G. Barbosa[2], R.A.F. Souza[5], A.O. Manzi[6], R. Seco[7], S. Sjostedt[8], J.-H. Park[9], A. Guenther[7,10], S. Kim[7], J. Smith[11,12], S.T. Martin[13,14] and M. Kalberer[1]*

[1]Department of Chemistry, University of Cambridge, Cambridge, CB2 1EW, UK

[2]Environmental Engineering Department, Federal University of Parana, Curitiba, Brazil

[3]School of Geography Earth & Environmental Sciences, University of Birmingham, Birmingham, B15 2TT, UK

[4]NCAS climate, University of Cambridge, Cambridge, CB2 1EW, UK

[5]State University of Amazonas, Av. Darcy Vergas, 1200, 69065-020, Manaus-AM, Brazil

[6]Instituto Nacional de Pesquisas da Amazônia (INPA), Clima e Ambiente (CLIAMB), Manaus-AM, Brazil

[7]Department of Earth System Science, University of California, Irvine CA 92697, USA

[8]NOAA ESRL Chemical Sciences Division, Boulder CO, USA.

[9]National Institute of Environmental Research, Republic of Korea

[10]Pacific Northwest National Laboratory, Richland WA, USA

[11]Atmospheric Chemistry Division, National Center for Atmospheric Research, Boulder CO, USA

[12]Dept of Chemistry, University of California, Irvine CA, USA

[13] Sch. Eng. & Appl. Sci., Harvard University, Cambridge, MA 02138 USA

[14]Dep. Earth & Planetary Sciences, Harvard University, Cambridge, MA 02138 USA

*Corresponding authors: I. Kourtchev (ink22@cam.ac.uk) and M. Kalberer (mk594@cam.ac.uk )

**Abstract**

The Amazon basin plays key role in atmospheric chemistry, biodiversity and climate change. In this study we applied nanoelectrospray (nanoESI) ultrahigh resolution mass spectrometry (UHR-MS) for the analysis of the organic fraction of $PM_{2.5}$ aerosol samples collected during dry and wet seasons at a site in central Amazonia receiving background air masses, biomass burning and urban pollution. Comprehensive mass spectral data evaluation methods (e.g., Kendrick Mass Defect, Van Krevelen diagrams, carbon oxidation state and aromaticity equivalent) were used to identify compound classes and mass distributions of the detected species. Nitrogen and/or sulfur containing organic species contributed up to 60% of the total identified number of formulae. A large number of molecular formulae in organic aerosol (OA) were attributed to later-generation nitrogen- and sulfur-containing oxidation products, suggesting that OA composition is affected by biomass burning and other, potentially anthropogenic, sources. Isoprene derived organo sulfate (IEPOX-OS) was found as the most dominant ion in most of the analysed samples and strongly followed the concentration trends of the gas-phase anthropogenic tracers confirming its mixed anthropogenic-biogenic origin. The presence of oxidised aromatic and nitro-aromatic compounds in the samples suggested a strong influence from biomass burning especially during the dry period. Aerosol samples from the dry period and under enhanced biomass burning conditions contained a large number of molecules with high carbon oxidation state and an increased number of aromatic compounds compared to that from wet. The results of this work demonstrate that the studied site is influenced not only by biogenic emissions from forest but also by biomass burning and potentially other anthropogenic emissions from the neighboring urban environments.

**Keywords:** organic aerosol, ultra-high resolution mass spectrometry, molecular composition, IEPOX-OS, Amazon.

## Introduction

The Amazon basin plays key role in atmospheric chemistry, biodiversity and climate change (Keller et al., 2009; Andrea et al., 2015). The Amazon rainforest is an important source of Biogenic Volatile Organic Compound (BVOC) emissions to the atmosphere (Greenberg et al., 2004; Alves et al., 2015), which give rise to secondary organic aerosol (SOA) through reaction with atmospheric oxidants (i.e. $O_3$, OH· and $NO_3^.$) (e.g., Martin et al., 2010). SOA particles scatter and absorb solar and terrestrial radiation, influence cloud formation, participate in chemical reactions in the atmosphere, and thus are suggested to play an important role in climate change (Andreae and Crutzen, 1997; Haywood and Boucher, 2000; Hallquist et al., 2009; Pöschl et al., 2010). Aerosol optical properties, which govern the ability to absorb solar radiation, strongly depend on SOA composition (Laskin et al., 2015). It has been shown that organic nitrates, nitrooxy-organosulfates and organic sulfates may contribute to light absorption by SOA (e.g., Song et al., 2013; Jacobson, 1999; Lu et al., 2011; Laskin et al., 2015). Chemical interactions between anthropogenic and biogenic aerosol precursors can play a significant role in the formation of SOA (Goldstein et al., 2009; Hoyle et al., 2011; Kleinman et al., 2015). For example, anthropogenic nitrogen oxides ($NO_x$) and sulfur dioxide ($SO_2$) are shown to react with a range of BVOCs leading to formation of organic nitrates (e.g., Roberts, 1990; Day et al., 2010; Fry et al., 2014), nitroxy-organosulfates and organosulfates (Surratt et al., 2008; Budisulistiorini et al., 2015). Much remains to be explored in terms of the molecular diversity of these compounds in the atmosphere.

A comprehensive knowledge of aerosol molecular composition, which in turn leads to better understanding of aerosol sources, is required for the development of effective air pollution mitigation strategies. However, identification of the organic aerosol composition, remains a major analytical challenge (Noziere et al., 2015). Organic aerosol is composed of thousands of organic compounds, which cover a wide range of physical and chemical properties (Goldstein and Galbally, 2007) making it difficult to find a single analytical technique for a detailed chemical analysis at the molecular level. Methods based on ultrahigh resolution mass

spectrometry (UHRMS) have shown great potential in solving this longstanding problem. UHR
mass spectrometers (e.g., Fourier transform ion cyclotron resonance MS and Orbitrap MS)
have mass resolution power that is at least one order of magnitude higher (≥100 000) than
conventional MS and high mass accuracy (<5 ppm) and thus, when coupled with soft
ionisation techniques (e.g., electrospray ionisation (ESI)), can provide a detailed molecular
composition of the organic aerosol (Nizkorodov et al., 2011, Noziere et al., 2015). Direct
infusion ESI-UHRMS has been applied successfully for the analysis of aerosol samples from
remote (e.g., boreal forest in Finland, Pico Island of the Azores archipelago), rural (e.g.,
Millbrook, USA; Harcum, USA; K-Puszta, Hungary) and urban (e.g., Cambridge, UK,
Birmingham, UK, Cork, Ireland, Shanghai, China and Los Angeles, USA) locations (Wozniak
et al., 2008; Schmitt-Kopplin et al., 2010; Kourtchev et al., 2013; 2014; Tao et al., 2014;
Dzepina et al., 2015).  UHRMS has proven to be extremely useful in assessing chemical
properties of the SOA.
The aim of this study was to investigate the detailed molecular composition of organic aerosol
from a site that received air masses from a wide range of origins, including the background
atmosphere of Amazonia, biomass burning and urban pollution plumes. The measurements
were performed as a part of the *Observations and Modeling of the Green Ocean Amazon*
(GoAmazon2014/5) campaign (Martin et al., 2016). The location of the research site where
aerosol was collected for this study is ~69 km downwind of Manaus (population 2 million),
intersected background and polluted air with day-to-day variability in the position of the
Manaus plume. The study designed served as a laboratory for investigating anthropogenic
perturbations to biogenic processes and atmospheric chemistry.
**Methods**
**Sampling site**
Aerosol sampling was conducted at site "T3" of GoAmazon2014/5 located at -3.2133º and -
60.5987º.º35'55 32'' W. The T3 site is located in the pasture area, ~2.5 km from the rainforest.

The air masses arriving to the sampling site often passed over the single large city (Manaus) in the region. Detailed descriptions of the site and instrumentation are provided in Martin et al. (2015).

PM$_{2.5}$ aerosol samples were collected on 47 mm polycarbonate filters Nuclepore, using a Harvard impactor (Air Diagnostics, Harrison, ME, EUA) with flow rate of 10 L min$^{-1}$ from 5 to 26 March 2014 and 5 Sept to 04 Oct of 2014, which were during Intensity Operating Periods 1 and 2 (IOP1 and IOP2) of GoAmazon2014/5, respectively, corresponding to the traditional periods of wet and dry seasons of Amazonia. The sampling durations are shown in the Table SI1. The airflow through the sampler was approximately 10 L min$^{-1}$ . After collection, the aerosol samples were transferred into Petri dishes and stored in the freezer at −4°C until analysis.

**Aerosol Sample Analysis**

Fifteen samples, 5 from IOP2 and 10 from IOP1, were extracted and analysed using a procedure described elsewhere (Kourtchev et al., 2014; Kourtchev et al., 2015). Depending on the aerosol loading of the analysed samples, a part (1/2 to whole) of the filter was extracted in methanol (Optima TM LC/MS grade, Fisher Scientific) in a chilled ice slurry, filtered through a Teflon filter (0.2 μm, ISODiscTM Supelco) and reduced by volume using a nitrogen line to achieve approximately 0.3 μg of aerosol per μL methanol. Several samples with the highest aerosol loading were divided into two parts for both direct infusion and LC/MS analyses while the samples with the lowest loading were only analysed using direct infusion analysis. The LC/MS portion was further evaporated to 20 μL and diluted to 100 μL by aqueous solution of formic acid (0.1%). The final extracts were analysed as described in Kourtchev et al. (2013) using a high-resolution LTQ Orbitrap Velos mass spectrometer (Thermo Fisher, Bremen, Germany) equipped with ESI and a TriVersa Nanomate robotic nanoflow chip-based ESI (Advion Biosciences, Ithaca NY, USA) sources. The Orbitrap MS was calibrated using an Ultramark 1621 solution (Sigma-Aldrich, UK). The mass accuracy of the instrument was below

1 ppm. The instrument mass resolution was 100 000 at $m/z$ 400. The ion transmission settings
were optimised using a mixture of camphor sulfonic acid (20 ng µL$^{-1}$) glutaric acid (30 ng µL$^{-}$
$^{1}$), and *cis*-pinonic acid (30 ng µL$^{-1}$) in methanol and Ultramark 1621 solution.

**Direct infusion UHRMS analysis**

The ionisation voltage and back pressure of the nanoESI direct infusion source were set at -
1.4 kV and 0.8 psi, respectively. The inlet temperature was 200 $^{0}$C and the sample flow rate
was approximately 200–300 nL min$^{-1}$. The negative ionisation mass spectra were collected in
three replicates at two mass ranges ($m/z$ 100–650 and $m/z$ 150–900) and processed using
Xcalibur 3.1 software (Thermo Fischer Scientific Inc.). Similar to our preceding studies
(Kourtchev et al., 2015) the average percentage of common peaks between analytical
replicates was ~80%. This is also in agreement with literature reports for similar data analysis
(Sleighter et al., 2012). The identification of IEPOX organosulfates was performed by
comparing MS fragmentation patterns and chromatographic elution time with a synthesised
IEPOX-OS standard which was provided by Dr Surratt from University of North Carolina. It
must be noted that due to competitive ionisation of analytes in the direct infusion ESI analysis
of the samples with a very complex matrix (i.e., aerosol extracts), the ion intensities do not
directly reflect the concentration of the molecules in the sample (Oss et al., 2010); therefore,
data shown in this work is semi-quantitative.

**LC-MS analysis**

LC-MS ESI parameters were as follows: spray voltage -3.6 kV; capillary temperature 300 $^{0}$C;
sheath gas flow 10 arbitrary units, auxiliary gas flow 10; sweep gas flow rate 5; S-lens RF level
58 %. LC/(-)ESI-MS analysis was performed using an Accela system (Thermo Scientific, San
Jose, USA) coupled with LTQ Orbitrap Velos MS and a T3 Atlantis C18 column (3 µm; 2.1 x
150 mm; Waters, Milford, USA). The sample extracts were injected at a flow rate of 200 µL
min$^{-1}$. The mobile phases consisted of 0.1% formic acid (v/v) (A) and methanol (B). The applied
gradient was as follows: 0–3 min 3% B, 3–25 min from 3 to 50% B (linear), 25–43 min from

50 to 90% B (linear), 43–48 min from 90 to 3% B (linear), and kept for 12 min at 3% B. The Collision Induced Dissociation (CID) settings for MS/MS analysis are reported in Kourtchev et al (2015).

**High resolution MS data analysis**

The direct infusion data analysis was performed using procedures described in detail by Kourtchev et al. (2013). Briefly, for each sample analysis, 60–90 mass spectral scans were averaged into one mass spectrum. Molecular formulae assignments were made using Xcalibur 3.1 software using the following constraints $^{12}C \leq 100$, $^{13}C \leq 1$, $^{1}H \leq 200$, $^{16}O \leq 50$, $^{14}N \leq 5$, $^{32}S \leq 2$, $^{34}S \leq 1$. The data processing was performed using a Mathematica 8.0 (Wolfram Research Inc., UK) code developed in-house that utilises a number of additional constraints described in previous studies (Kourtchev et al., 2013; Kourtchev et al., 2015). Only ions that appeared in all three replicates were kept for evaluation. The background spectra obtained from the procedural blanks were also processed using the rules mentioned above. The formulae lists of the background spectra were subtracted from those of the ambient (or chamber) sample and only formulae with a sample/blank peak intensity ratio $\geq 10$ were retained

The Kendrick Mass Defect (KMD) is calculated from the difference between the nominal mass of the molecule and the exact KM (Kendrick, 1963). Kendrick mass of the $CH_2$ unit is calculated by renormalising the exact IUPAC mass of $CH_2$ (14.01565) to 14.00000.

**Benzene and isoprene measurements**

For benzene and isoprene analysis we used a high-resolution selective-reagent-ionisation proton transfer reaction time-of-flight mass spectrometer (SRI-PTR-TOF-MS 8000, Ionicon Analytik, Austria). A description of the PTR-TOF-MS instrument and the data reduction process used are provided elsewhere (Graus et al. 2010; Müller et al. 2013). Background of the instrument was measured regularly by passing ambient air through a platinum catalyst heated to 380 ºC. Sensitivity calibrations were performed by dynamic dilution of VOCs using

several multi-component gas standards (Apel Riemer Environmental Inc., Scott-Marrin, and Air Liquide, USA). The calibration cylinders contained acetaldehyde, acetone, benzene, isoprene, $\alpha$-pinene, toluene and trichlorobenzene, among others. During IOP1, the instrument was operated with $H_3O^+$ reagent ion and at a drift tube pressure of 2.3 mbar, voltage of 600 V, and temperature of 60 $^oC$, corresponding to a a field density ratio E/N ratio of 130 Td (E being the electric field strength and N the gas number density; 1 Td = $10^{-17}$ V $cm^{-2}$). During IOP2, the reagent ion was $NO^+$ and the drift tube settings were 2.3 mbar, 350 V, and 60 $^oC$, resulting in an E/N ratio of 76 Td. The sampling was done with 1 min time resolution and the instrument detection limit for benzene and isoprene were below 0.02 and 0.04 ppbv, respectively.

**Air mass history analysis**

Air mass history analysis was done for the sampling period using the Numerical Atmospheric-dispersion Modeling Environment (NAME) model, developed by the UK Met Office (Maryon et al., 1991). NAME is a Lagrangian model in which particles are released into 3D wind fields from the operational output of the UK Met Office Unified Model meteorology data (Davies et al., 2005). These winds have a horizontal resolution of 17 km and 70 vertical levels up that reach ~80 km. In addition, a random walk technique was used to model the effects of turbulence on the trajectories (Ryall and Maryon, 1998). To allow the calculation of air mass history for the average sampling time (which varied between samples, 24, 36 or 48 hours), 10 000 particles per hour were released continuously from the T3 site. The trajectories travelled back in time for 3 days with the position of the particles in the lowest 100 m of the model atmosphere recorded every 15 min. The particle mass below 100 m was integrated over the 72 h travel time. The air mass history ('footprints') for the periods of the analysed filters are shown in Figure SI1. The majority of the three-day air mass footprints originated from the east, although wind direction showed variability nearer to the sampling site on some occasions e.g., sample MP14-17 (Fig. SI1). Almost all air masses pass over Manaus and therefore highlight this city as a potential source. Some air masses also pass over Manacapuru, but this is rare

and the corresponding time-integrated concentrations are lower than the equivalent Manaus
values.

**Results and discussions**

Figure 1 shows mass spectra from two typical samples collected during IOP1 and IOP2. The
majority of the ions were associated with molecules below 500 Da although the measured
mass goes up to 900 Da. Although ESI is a 'soft' ionisation technique resulting in minimal
fragmentation, we cannot exclude the possibility that some of the detected ions correspond to
fragments, also in light of the many relative fragile compounds (e.g., highly oxygenated
compounds) that constitute OA. The largest group of identified molecular formulae in all
samples were attributed to molecules containing CHO atoms only (1051±141 formulae during
IOP2 and 820±139 during IOP1), followed by CHON (537±71 during IOP2 and 329±71 during
IOP1), CHOS (183±34 during IOP2 and 137±31 during IOP1) and CHONS (37±11 during
IOP2 and 28±10 during IOP1) (Fig. 2).The number of molecular formulae containing CHO and
CHON subgroups increased by ~20% from IOP1 to IOP2 period; however, rather insignificant
increase was observed for CHOS and CHONS subgroups. The Student's t-test showed that
the observed difference for CHO (p=0.0092) and CHON (p=0.00007) subgroups between two
seasons is statistically significant. This is consistent with the observed increase in odd reactive
nitrogen species ($NO_y$) from IOP1 to IOP2 (Table SI1). Organic nitrates are believed to form
in polluted air through reaction with nitrogen oxides during day and from reaction of $NO_3^{.}$ with
BVOCs during nighttime (Day et al., 2010; Ayres et al., 2015). The average concentration of
$NO_y$ during IOP1 was found to be on almost two times higher, which is possibly reflected in
the increased number of organonitrates in the aerosol samples from IOP2. Moreover, the
increase in the number of organonitrates during IOP2 is consistent with the recent studies,
which demonstrated that organonitrates groups in aerosol particles may hydrolyse under high
RH conditions (Liu et al., 2012). In this respect, while night time maximum RH during both filter
sampling periods was very similar (~90%), day-time RH during IOP1 was higher (89%)
compared to that from the IOP2 period (66%) (Fig. SI2).
Carbon oxidation state ($OS_C$) introduced by Kroll et al. (2011) can be used to describe the
composition of a complex mixture of organics undergoing oxidation processes. $OS_C$ was
calculated for each molecular formula identified in the mass spectra using the following
equation:
$$OS_C = -\sum_i OS_i \frac{n_i}{n_C} \qquad \text{(Eq. 1)}$$
where $OS_i$ is the oxidation state associated with element i, $n_i/n_C$ is the molar ratio of element
i to carbon within the molecule (Kroll et al., 2011).
Figure 3 shows overlaid OSc plots for two samples from IOP1 and IOP2. Consistent with
previous studies, the majority of molecules in the sampled organic aerosol had $OS_C$ between
−1.5 and +1 with up to 30 (nC) carbon atoms throughout the selected mass range (*m/z* 100-
650) (Kroll et al., 2011 and the references therein). The molecules with $OS_C$ between –1 and
+1 with 13 or less carbon atoms (nC) are suggested to be associated with semivolatile and
low-volatility oxidised organic aerosol (SV-OOA and LV-OOA) produced by multistep oxidation
reactions. The molecules with OSc between -0.5 and -1.5 with 7 or more carbon atoms are
associated with primary biomass burning organic aerosol (BBOA) directly emitted into the
atmosphere (Kroll et al., 2011). The cluster of molecules with $OS_C$ between –1 and –1.5 and
nC less than 10 could be possibly associated with OH radical oxidation products of isoprene
(Kourtchev et al., 2015), which is an abundant VOC in Amazon rain forest (Rasmussen and
Khalil, 1988; Chen et al., 2015).  The isoprene daytime average was above 1.5 ppbv during
both seasons, with hourly campaign-averages reaching up to 2.3 and 3.4 ppbv for IOP1 and
IOP2, respectively. In general, aerosol samples from IOP1 contained less oxidised molecules
compared to those from IOP2. Wet deposition of aged or processed aerosol during wet (i.e.,
IOP 2) sampling period cannot be the only reason for the observed differences in OSc. It has
been shown that different oxidation regimes to generate SOA (e.g., OH radical vs. ozonolysis)
can result in significantly different OSc of SOA (Kourtchev et al., 2015). For example, the SOA
component from OH initiated oxidation of α-pinene as well as BVOC mixtures had a molecular
composition with higher OSc throughout the entire molecular mass range compared to that
obtained from ozonolysis reaction (Kourtchev et al., 2015).
Figure 4 shows the distribution of ion intensities for selected tentatively identified tracer
compounds for anthropogenic, biogenic and mixed sources in all 15 samples. The structural
or isomeric information is not directly obtained from the direct infusion analysis; therefore, the
identification of the tracer compounds was achieved by comparing MS/MS fragmentation
patterns from authentic standards and published literature. The tracer compounds include
anhydrosugars, structural isomers with a molecular formula $C_6H_{10}O_5$ at $m/z$ 161.0456
corresponding to levoglucosan, mannosan, galactosan and 1,6-anhydro-$β$-$D$-glucofuranose,
which are regarded as marker compounds for biomass burning (Simoneit et al., 1999;
Pashynska et al., 2002; Kourtchev et al., 2011). Nitrocatechols, with a molecular formula
$C_6H_5NO_4$ ($m/z$ 154.01458) are attributed to mixed anthropogenic sources, e.g., biomass and
vehicular emissions and methyl-nitrocatechols ($C_7H_7NO_4$, $m/z$ 168.03023) are important
markers for biomass burning OA, formed from $m$-cresol emitted during biomass burning
(Iinuma et al., 2010). 3-methyl-1,2,3-butanetricarboxylic acid (3-MBTCA), with a molecular
formula $C_8H_{12}O_6$ at $m/z$ 203.05611, is an OH-initiated oxidation product of $α$- and $β$-pinene
(Szmigielski et al., 2007), and regarded as a tracer for processed or biogenic SOA. Finally,
isoprene epoxydiol organosulfate ester (IEPOX-OS), with a molecular formula $C_5H_{12}O_7S$ at
$m/z$ 215.0231, is shown in Figure 4. From studies in mid latitude environments it has been
suggested that IEPOX-OS is formed through reactions between $SO_x$ and isoprene oxidation
products (Pye et al., 2013; Budisulistiorini et al., 2015) and thus can be used to observe the
extent of $SO_2$ aging effects on the biogenic SOA. Direct infusion analysis suffers from
competitive ionisation in the complex matrices and thus comparing ion intensities across
samples has to be done with caution. Moreover, other compounds with similar molecular
composition present in the aerosol matrix may also contribute to the ion intensities of the
discussed above compounds. All selected tracers showed very similar variations with benzene
concentration that was measured in the gas-phase using PTR-MS (Fig. 3). Benzene, generally
regarded as an anthropogenic species, has various sources including industrial solvent
production, vehicular emissions and biomass burning (Hsieh et al., 1999; Seco et al., 2013;
Friedli et al., 2001). Recent studies indicated that vegetation (leaves, flowers, and
phytoplankton) emits a wide variety of benzenoid compounds to the atmosphere at substantial
rates (Misztal et al., 2015). However, considering that benzene concentration correlated very
well with another anthropogenic tracer CO ($R^2$=0.77, Figure SI3) during IOP1 and IOP2
periods, it is rather likely that the observed benzene concentrations were mainly due to
anthropogenic emissions. During the sampling period, irrespectively of the season, air masses
passed over the large city Manaus and small municipalities located near the T3 site (Figure
SI1). It must be noted that due to rather low sampling resolution time (≥24h) the molecular
composition of all analysed samples is likely to be influenced by clean air masses and
anthropogenic plumes from these urban locations which usually last only a few hours per day
and thus individual urban plume events cannot be identified with the data analysed here. In
Manaus natural gas is mainly used for heating and cooking and therefore, the contribution
from these activities to biomass burning OA at our site is highly unlikely. During IOP1 much
lower incidents of forest fires were observed compared to that during IOP2 (Martin et al.,
2016). For example, a number of forest fires in the radius of 200 km from the sampling site
varied between 0 to 340 fires (http://www.dpi.inpe.br/proarco/bdqueimadas/). This is reflected
in the ion intensities of the particle phase biomass burning markers, i.e., anhydrosugars
($C_6H_{10}O_5$) and nitrocatechols ($C_6H_5NO_4$) and gas-phase benzene concentrations, which were
significantly lower during IOP1 compared to that from IOP2, when on average more fires are
observed.
It should be noted that ion intensities for anhydrosugars ($C_6H_{10}O_5$) and nitrocatechols
($C_6H_5NO_4$) showed very good correlation ($R^2$>0.7) suggesting that nitrocatechols, observed at
the sampling site, are mainly associated with biomass burning sources. The highest ion
intensities of these tracer compounds were observed during two periods: 7-9 September 2014
(sample MP14-128) and 27-28 September 2014 (sample MP14-148) with the later one
coinciding with highest incident of fires (340 fires). Although during 7-9 September (sample
MP14-128) a significantly lower number (22 fires) of fires was observed compared to the
period of 27-28 September 2014, lower wind speed occurring during 7-9 September suggests
that high intensity of the biomass burning markers could be due to the biomass burning
emissions from nearby sources. Between the T3 sampling site and Manaus (about 20 km east
of the site), there are a number of small brick factories, which use wood to fire the kilns (Martin
et al., 2016) and thus they are an additional local wood burning source besides the forest and
pasture fires.
Interestingly the sample MP14-148 had the highest ion intensity corresponding to IEPOX-OS
(Fig. 4), which also coincided with the strong increase of the ion intensity at $m/z$ 96.95987
corresponding to $[HSO_4]^-$. This is consistent with organosulfates formation mechanism through
reactive uptake of isoprene epoxydiols (IEPOX) in the presence of acidic sulfate seed (Surratt
et al., 2010; Lin et al., 2012; 2013). A similar relationship between sulfate and organosulfates
concentrations has been observed previously in field studies in the Southeastern US (Surratt
et al., 2007, 2008, 2010; Lin et al., 2012, 2013). This is also in agreement with previous studies
from Amazon where the highest levels of 2-methyltetrols were observed during the dry period
which was characterised by biomass burning (and higher particle concentrations of sulfuric
acid) (Claeys et al., 2010). Considering that Claeys et al (2010) employed alternative GC/MS
procedure with prior trimethylsylilation, 2-methyltetrol sulfates were converted to 2-
methyltetrols and not detectable as separate OS compounds. It should be noted that the 27-
28 September period (sample MP14-148) was marked by a very strong increase in the CO
concentration (Fig. SI4). In mid-latitude environments it has been suggested that the
production of anthropogenic SOA in an air mass, as it travels from an urban source region,
can be estimated by using a relatively inert pollution tracer, such as CO occurring in the air
mass (De Gouw et al., 2005; Hoyle et al., 2011). At T3 sampling site, highest CO
concentrations are observed in air masses affected by biomass burning. Therefore, it is
possible that organic aerosol in the sample MP14-148 has experienced the highest
contribution from biomass burning as well as other anthropogenic activities.
To investigate the influence of anthropogenic activities (i.e., biomass burning) on a detailed
molecular composition of organic aerosol at the T3 site we compared samples from the
periods with the lowest (9 fires), moderately high (254 fires) and the highest (340 fires)
incidents of fires occurring within 200 km around the site.
Figure 5 (a-c) shows H/C ratios of CHO containing formulae as a function of their molecular
mass and double bond equivalent (DBE), which shows a degree of unsaturation of the
molecule, for a sample with the lowest (a) moderately high (b) and highest incidents (c) of
fires. One of the obvious differences between these samples is the abundance of ions with
low H/C ratios (< 1). The majority of these ions have DBE above 7 indicating that they likely
correspond to oxidised aromatic compounds, which are mainly of anthropogenic origin
(Kourtchev et al., 2014; Tong et al., 2016). For example, the smallest polycyclic aromatic
hydrocarbon (PAH), naphthalene with a molecular formulae $C_{10}H_8$ has an H/C=0.8 and
DBE=7. The number of CHO containing formulae with high DBE equivalent and low H/C
increased dramatically during the days with moderately high and high incidents of fires (Fig.
5a-c), suggesting that they are mainly associated with biomass burning. The largest grey
circles in Fig 5(a-c) correspond to the ions at $m/z$ 133.01425 (with neutral molecular formula
$C_4H_6O_5$), $m/z$ 187.0612 ($C_8H_{12}O_5$), $m/z$ 201.07685 ($C_9H_{14}O_5$), $m/z$ 203.05611 ($C_8H_{12}O_6$), and
$m/z$ 215.05611 ($C_9H_{12}O_6$) with DBE<6.
Recent studies indicated that different families of compounds with heteroatoms (e.g. O, S)
overlap in terms of DBE and thus may not accurately indicate the level of unsaturation of
organic compounds. For example, the divalent atoms, such as oxygen and sulphur, do not
influence the value of DBE, yet they may contribute to the potential double bonds of that
molecule (Reemtsma 2009; Yassine et al., 2014). Yassine et al (2014) suggested using
aromaticity equivalent ($X_c$), to improve the identification and characterisation of aromatic and
condensed aromatic compounds in WSOC. The aromaticity equivalent can be calculated as
follows:
$$X_c = \frac{3(DBE - (mN_O + nN_S)) - 2}{DBE - (mN_O + nN_S)}$$          (Eq. 2)
where 'm' and 'n' correspond to a fraction of oxygen and sulfur atoms involved in π–bond
structures of a compound, which varies depending on the compound class. For example,
carboxylic acids, esters, and nitro functional groups have m=n=0.5. For compounds containing
functional groups such as aldehydes, ketones, nitroso, cyanate, alcohol, or ethers 'm' and 'n'
are 1 or 0. Considering that ESI, in negative mode, is most sensitive to compounds containing
carboxylic groups we, therefore, used m=n=0.5 for the calculation of the Xc. For molecular
formulae with an odd number of oxygen or sulfur, the sum $(mN_O + nN_S)$ in Eq. 2 was rounded
down to the closest integer as detailed in Yassine et al (2014). The authors proposed that
aromaticity equivalent with Xc ≥2.50 and Xc ≥2.71 as unambiguous minimum criteria for the
presence of aromatics and condensed aromatics.
Expressing our data using aromaticity equivalents confirmed that the increase in the number
of molecules with high DBE from the sample with the lowest to the highest incidents of fires
was due to the increase in the number of aromatic and condensed aromatic compounds in the
aerosol samples (Figures SI5).  Considering the Yassine et al. (2014) assignment criteria for
the aromatic-reach matrices, the highest number of the aromatic compounds in the Amazon
samples was observed for formulae with a benzene core structure (Xc =2.50) followed by
formulae with pyrene core structure (Xc = 2.83), and an ovalene core structure (Xc =2.92) as
well as highly condensed aromatic structures or highly unsaturated compounds (Xc >2.93).
The largest grey circles in Figure SI5a correspond to the ions at *m/z* 187.11357 with a neutral
molecular formula $C_9H_{17}NO_3$ and *m/z* 281.26459 with a neutral molecular formula $C_{18}H_{35}NO$.
The largest grey circles in Figure SI5b and c correspond to the ions at *m/z* 154.0146, *m/z*
168.03023 and *m/z* 152.03532 with neutral molecular formulae $C_6H_5NO_4$, $C_7H_7NO_4$ and
$C_7H_7NO_3$, respectively.
Interestingly, a similar trend was observed for the molecules containing CHON subgroups
(Figure SI6). A number of CHON molecules with low H/C (<1) and high DBE (≥5) almost
doubled from the days with 9 to 340 fires (Figure SI7). Nitro-aromatic compounds, such as
nitrophenols (DBE=5) and N-heterocyclic compounds including 4-nitrocatechol and isomeric
methyl-nitrocatechols are often observed in the OA from the biomass burning sources
(Kitanovski et al., 2012a,b; Iinuma et al., 2010) and have been  suggested as potential
contributors to light absorption by brown carbon (Laskin et al., 2015). It is worth mentioning
that aerosol samples affected by biomass burning contained another interesting ion at *m/z*
182.04588 with a neutral molecular formula $C_8H_9NO_4$, possibly corresponding to biomass
burning OA markers isomeric dimethyl-nitrocatechols (Kahnt et al., 2013). The differences in
the increased number of nitro-artomatic compounds in aerosol samples affected by biomass
burning are also apparent in overlaid Van Krevelen diagrams (Figure 6), which show H/C and
O/C ratios for each formula in a sample. Van Krevelen diagrams, can be used to describe the
overall composition or evolution of organic mixtures (Van Krevelen, 1993; Nizkorodov et al.,
2011; Noziere et al., 2015). Organic aerosol affected by biomass burning contained
significantly larger number of CHON formulae with O/C < 0.5 and H/C < 1 (Fig. 6a and b, area
B) but smaller number of formulae with O/C < 0.5 and H/C > 1. (Fig. 6a and b, area A). While
molecules with H/C ratios (<1.0) and O/C ratios (<0.5) (area A in Fig. 3) are generally
associated with aliphatic compounds typically belong to oxidised aromatic hydrocarbons,
molecules with high H/C ratios (>1.5) and low O/C ratios (<0.5) (area B in Fig. 3) (Mazzoleni
et al., 2010; 2012).  Although the smaller number of nitro-aliphatic compounds in the samples
affected by biomass burning requires further investigation, it is possible that they were oxidised
in the polluted air by $NO_x$ and $O_3$ (Zahardis et al., 2009; Malloy et al., 2009), which production
is significantly enhanced during fire events (e.g., Galanter et al., 2000). The majority (up to
80%) of the CHON molecules in the analysed samples have O/C ratios < 0.7 (Fig. 6). The
relatively low oxygen content suggests that these molecules include decreased nitrogen-
containing compounds (Zhao et al, 2013). Although biomass burning material type is expected
to result in different molecular composition, the presence of a large number of molecules with
low O/C ratio is consistent with the literature. For example, most of the CHON molecules in
OA from wheat straw burning in K-puszta, Great Hungarian Plain in Hungary and biomass
burning at Canadian rural sites (Saint Anicet, Quebec, and Canterbury, New Brunswick) had
O/C ratios below 0.7 (Schmitt-Kopplin et al., 2010). In addition, the CHON molecules identified
by LC/MS in biomass burning OA from Amazonia showed O/C ratios below 0.7, i.e., 4-
nitrocatechol ($C_6H_5NO_4$; O/C = 0.67), isomeric methyl-nitrocatechols ($C_7H_7NO_4$; O/C = 0.57),
and isomeric dimethyl-nitrocatechols ($C_8H_9NO_4$; O/C = 0.50) (Claeys et al., 2012).
Figure 7 shows overlaid OSc plots for OA from the days with low, moderately high and high
incidents of fires.  During the days affected by high and moderately high number of fires, OSc
was shifted towards more oxidised state for the CHO molecules containing more than 7 carbon
atoms. The difference in OSc becomes even more pronounced with the increased number of
carbons (e.g. >7 carbon atoms) in the detected molecular formulae.  Interestingly, the affected
ions with high OSc do not fall into the category of the BBOA (encircled area in Fig. 7) which
are associated with primary particulate matter directly emitted into the atmosphere as defined
in Kroll et al (2011).
At first glance, biomass burning seems to influence the number and intensity of the CHOS
containing formulae; however, the effect was at a much lower extent compared to that for the
CHO and CHON molecules (see discussion above). Higher number of CHOS containing
molecules was observed in the sample (MP14-148) corresponding to the highest incident of
fires (Figures 8a). Interestingly, IEPOX-OS was found to be very abundant in the sample that
experienced the highest incidents of fires (Figure 8a). The significant IEPOX-OS mass was
previously observed during a low-altitude flight campaigns at Northern California and southern
Oregon at high NO conditions (> 500 pptv) (Liao et al., 2015). The authors explained this
observation by the transport or formation of IEPOX from isoprene hydroxynitrate oxidation
(Jacobs et al., 2014) and higher sulphate aerosol concentrations occurring during their
sampling period (Nguyen et al., 2014). This explanation is also consistent with our results. The
ion at *m/z* 96.95987 corresponding $[HSO_4]^-$ in UHR mass spectra of the sample MP14-148
was three times more abundant compared that in the sample MP14-129 suggesting that
particle acidity may be one of the reasons for the high abundance of the IEPOX-OS in this
sample. Considering that the main sources of sulphate at T3 site are industrial pollution (e.g.,
power plants), natural and long range-sources, they could also be responsible for the high
abundance of the sulphate and IEPOX-OS in the samples besides the overlapping biomass
burning event. Noticeably, these samples not only contained a larger number of oxygenated
CHOS-containing molecules with O/C>1.2 but also molecules with O/C<0.6 and H/C ranging
from 0.4 to 2.2. Recent laboratory and field studies indicated the presence of a large number
of aromatic and aliphatic OSs and sulfonates in OA and linked them to anthropogenic
precursors (Tao et al., 2014; Wang et al., 2015; Riva et al., 2015; 2016; Kuang et al., 2016).
Riva et al (2015, 2016) demonstrated formation of OSs and sulfonates in the laboratory smog
chamber experiments from photooxidation of alkanes and PAHs, respectively. The authors
indicated enhancement of organosulfates yields in the presence of the acidified ammonium
sulphate seed and suggested that these OSs are mainly formed through reactive uptake of
gas-phase epoxides. It must be noted that above cited field studies are based on
measurements at the Northern Hemisphere USA and thus organosulfates formation pathways
and sources may differ from that of Amazonia.
KMD plots are useful visualisation technique for identification of homologous series of
compounds differing only by the number of a specific base unit (e.g., a $CH_2$ group).
Anthropogenically affected aerosol samples have longer homologous series of molecules
containing CHOS subgroups (Figure 8b).  One of these longer series includes a second most
intensive ion at $m/z$ 213.0075 ($C_5H_{10}O_7S$). The compound with molecular formula $C_5H_{10}O_7S$
has been previously observed in the laboratory and field studies and attributed to isoprene
derived organosulfates (Surratt et al., 2008; Gómez-González, 2008; Kristensen and Glassius,
2011; Nguyen et al., 2014; Hettiyadura et al., 2015). This molecular formula could also be
associated with organosulfates (e.g., isomeric 3-sulfooxy-2-hydroxypentanoic acid and 2-
sulfooxy-3-hydroxypentanoic acid) formed from the green leaf volatiles 2-*E*-pentenal, 2-*E*-
hexenal, and 3-hexenal (Shalamzari et al., 2016). The KMD plot (Figure 8b) shows that OA
from the anthropogenically affected samples contained an additional series of CHOS
molecules with high KMD >0.33 that were not present in the sample from the less polluted
period.  Most of these ions are highly oxygenated (containing >10 oxygens) and are likely to
be associate with molecules produced through photochemical ageing reactions (Hildebrandt
et al., 2010).
It is worth noting that in the most of the samples IEPOX-OS was not a part of any homologous
series in KMD plot (e.g., Fig 8b). This observation confirms that atmospheric oxidation
reactions resulting in the incorporation of S and N functional groups do not always conserve
homologous series but could also lead to a wide range of possible reaction products (Rincon
et al., 2012; Kourtchev et al., 2013).
**Conclusions**
In this study we applied direct infusion nanoESI UHR-MS for the analysis of the organic
fraction of $PM_{2.5}$ samples collected IOP1 and IOP2 of GoAmazon2014/5 in central Amazonia
which is influenced by both background and polluted air masses. Up to 2100 elemental
formulae were identified in the samples, with the largest number of formulae found during
IOP2. The distribution of several tracer compounds along with the comprehensive mass
spectral data evaluation methods (e.g., Kendrick Mass Defect, Van Krevelen diagrams,
carbon oxidation state and aromaticity equivalent) applied to the large UHRMS datasets were
used to identify various sources of organic aerosol components, including natural biogenic
sources, biomass burning and anthropogenic emissions. The distinguishable homologous
series in the KMD diagram contained nitrogen-containing series included NACs, e.g.,
nitrocatechols, nitrophenols, nitroguaiacols and nitrosalicylic acids derived from biomass
burning material. Isoprene derived IEPOX-OS was found as the most dominant ion in most of
the analysed samples and strongly followed the concentration trends of the gas-phase
anthropogenic tracer benzene and CO (with biomass burning as dominant tracer at the T3

site) supporting its mixed biomass burning-anthropogenic-biogenic origin. Van Krevelen, DBE and Xc distributions along with relatively low elemental O/C and H/C ratios indicated the presence of a large number of oxidised aromatic compounds in the samples. A significant number of CHO containing formulae in aerosol samples from IOP2 had higher oxidation state compared to that from IOP1 and became even more important during the days with the highest incidents of fires. Although our results suggest that the studied site is not only significantly influenced by biogenic emissions and biomass burning but also anthropogenic emissions from the neighboring urban activities, future work is needed to better understand the quantitative contributions of the various factors to the aerosol composition at the T3 site. The analysis of aerosol samples with higher sampling resolution or quantifying specific marker compounds and applying a receptor modelling techniques (Alves et al., 2015) would allow separating these sources in more detail and thus significantly improve understanding of the aerosol formation sources at the site.

**Acknowledgment:**

Research at the University of Cambridge was supported by the ERC grant no. 279405. The authors would like to thank Dr Jason Surratt (University of North Carolina) for providing a synthesised IEPOX-OS standard. $O_3$, CO, $NO_y$, RH and rain data were obtained from the Atmospheric Radiation Measurement (ARM) Climate Research Facility, a U.S. Department of Energy Office of Science user facility sponsored by the Office of Biological and Environmental Research. We acknowledge the support from the Central Office of the Large Scale Biosphere Atmosphere Experiment in Amazonia (LBA), the Instituto Nacional de Pesquisas da Amazonia (INPA), and the Universidade do Estado do Amazonia (UEA). The work was conducted under 001030/2012-4 of the Brazilian National Council for Scientific and Technological Development (CNPq).

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

R.M., and Kalberer, M.: FDATMOS16 Molecular Composition of Organic Aerosols at Urban
Background and Road Tunnel sites using Ultra-high Resolution Mass Spectrometry, Faraday
Discuss., DOI: 10.1039/C5FD00206K, 2016.
Wang, X.K., Rossignol, S., Ma, Y, Yao, L., Wang, M.Y., Chen, J.M., George, C., and Wang,
L.: Identification of particulate organosulfates in three megacities at the middle and lower
reaches of the Yangtze River, Atmos. Chem. Phys. Discuss., 15, 21414-21448, 2015.
Wozniak, A. S., Bauer, J. E., Sleighter, R. L., Dickhut, R. M., and Hatcher, P. G.: Technical
Note: Molecular characterization of aerosol-derived water soluble organic carbon using
ultrahigh resolution electrospray ionization Fourier transform ion cyclotron resonance mass
spectrometry, Atmos. Chem. Phys., 8, 5099– 5111, 2008.
Yassine, M. M., Harir, M., Dabek-Zlotorzynska, E., and Schmitt-Kopplin, P: Structural
characterization of organic aerosol using Fourier transform ion cyclotron resonance mass
spectrometry: aromaticity equivalent approach. Rapid Commun. Mass Sp., 28, 2445-2454,
847     2014.

Zahardis, J., Geddes, S., and Petrucci, G. A.: The ozonolysis of primary aliphatic amines in
fine particles, Atmos. Chem. Phys., 8, 1181-1194, 2008.
Zhao, Y., Hallar, A. G., and Mazzoleni, L. R.: Atmospheric organic matter in clouds: exact
masses and molecular formula identification using ultrahigh-resolution FT-ICR mass
spectrometry, Atmos. Chem. Phys., 13, 12343-12362, 2013.




**Figures:**

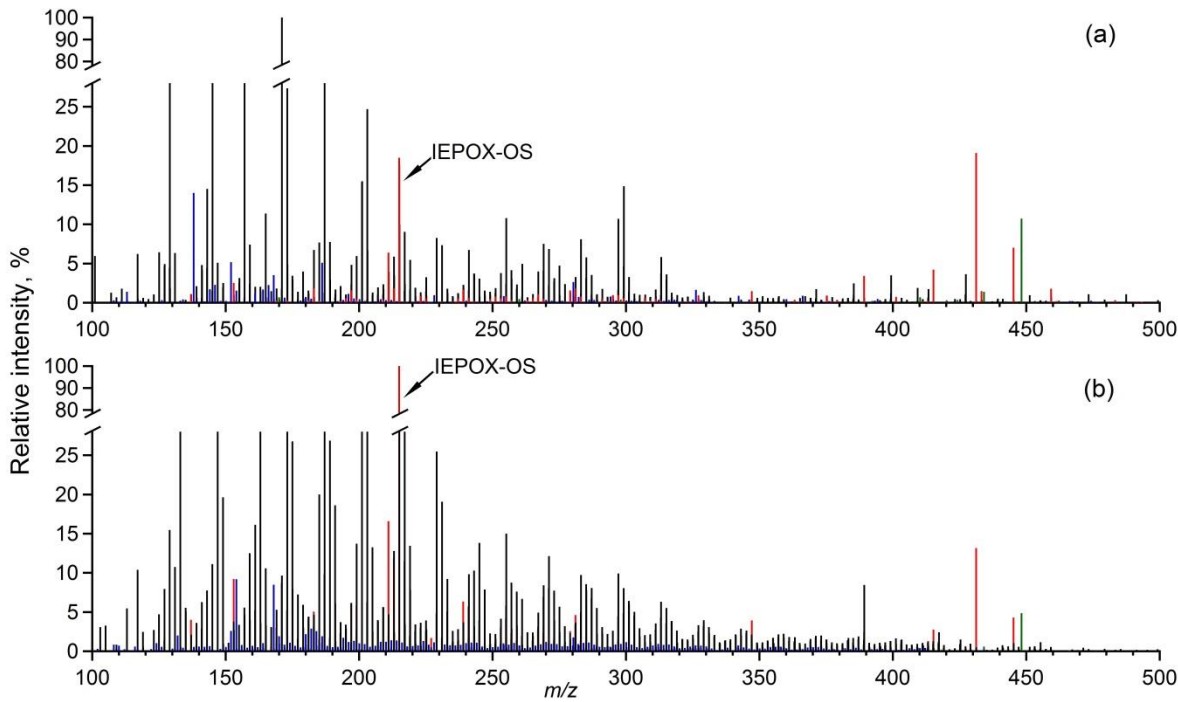


Figure 1. (-)-nanoESI-UHRMS of the representative PM2.5 samples during (a) IOP1 (b) IOP2. The line colours in the mass spectra correspond to the CHO (black), blue (CHON), CHOS (red) and CHONS (green) formulae assignments. The relative intensity axis was split to make a large number of ions with low intensities visible.

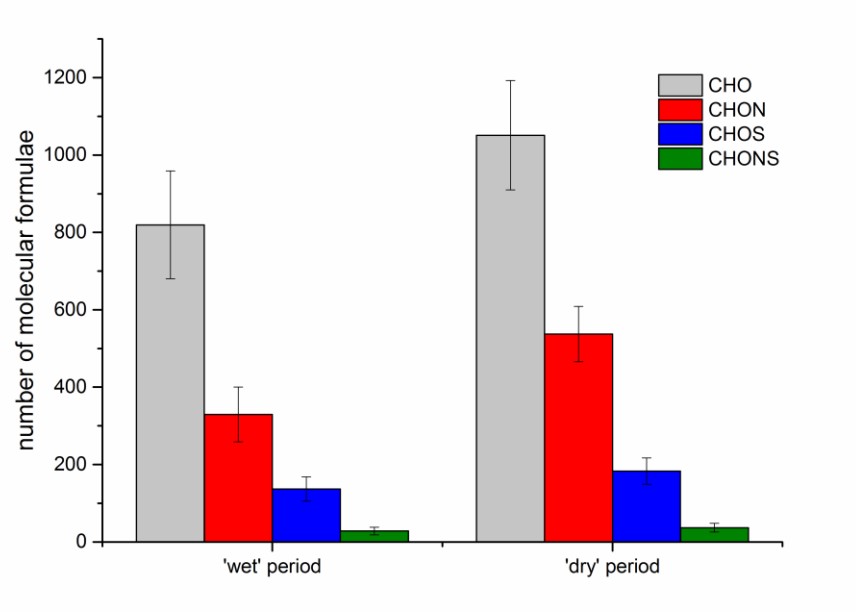

863

Figure 2. Average number of molecular formulae during IOP1 and IOP2. Standard deviation bars show variations between samples within individual season.

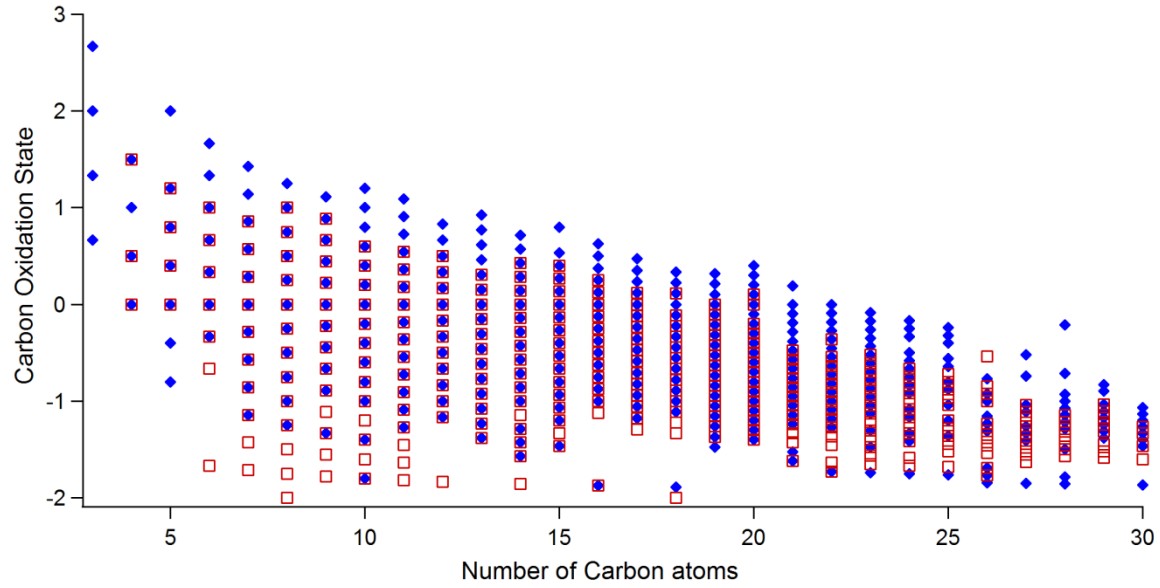

866

Figure 3. Carbon oxidation state plot for CHO containing formulae in organic aerosol from IOP1 (red squares) and IOP2 (blue diamonds).

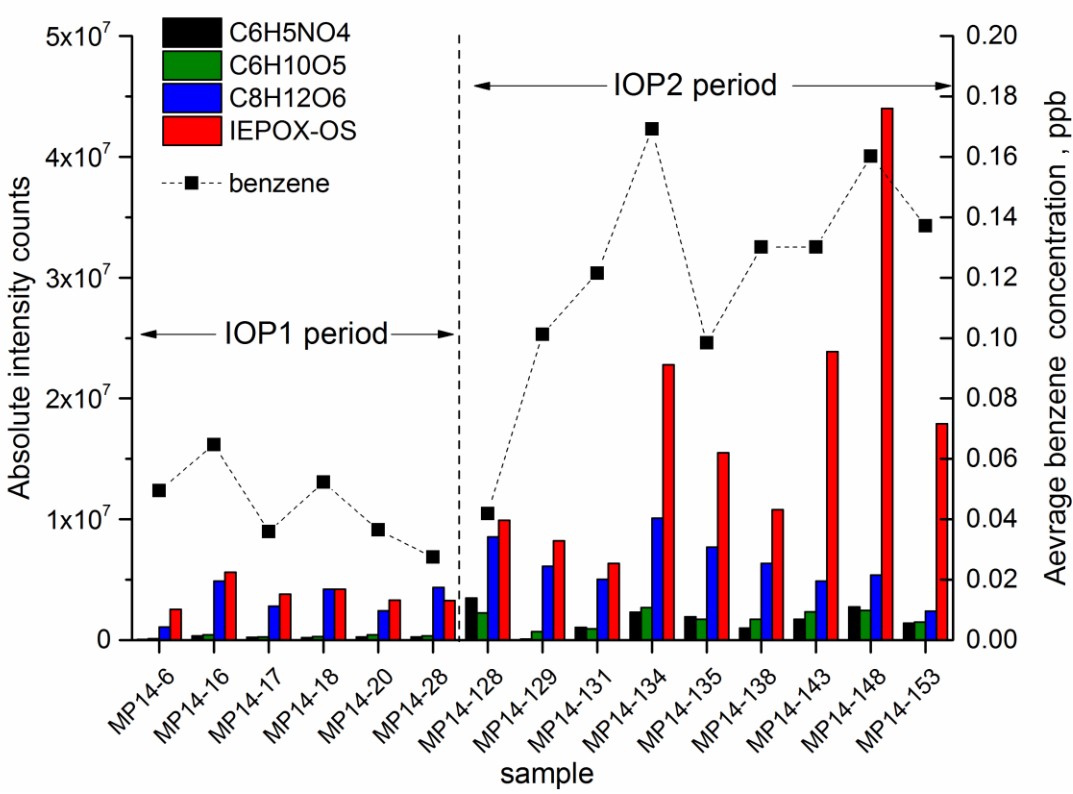

869

Figure 4. Ion intensity distributions (left axis) of selected tentatively identified markers in individual samples using UHRMS analysis and averaged benzene concentration (right axis) from PTR-TOF-MS analysis. Benzene concentration was averaged for the aerosol filter sampling intervals. The UHRMS data was corrected for organic carbon load in each individual filter sample (see method section).

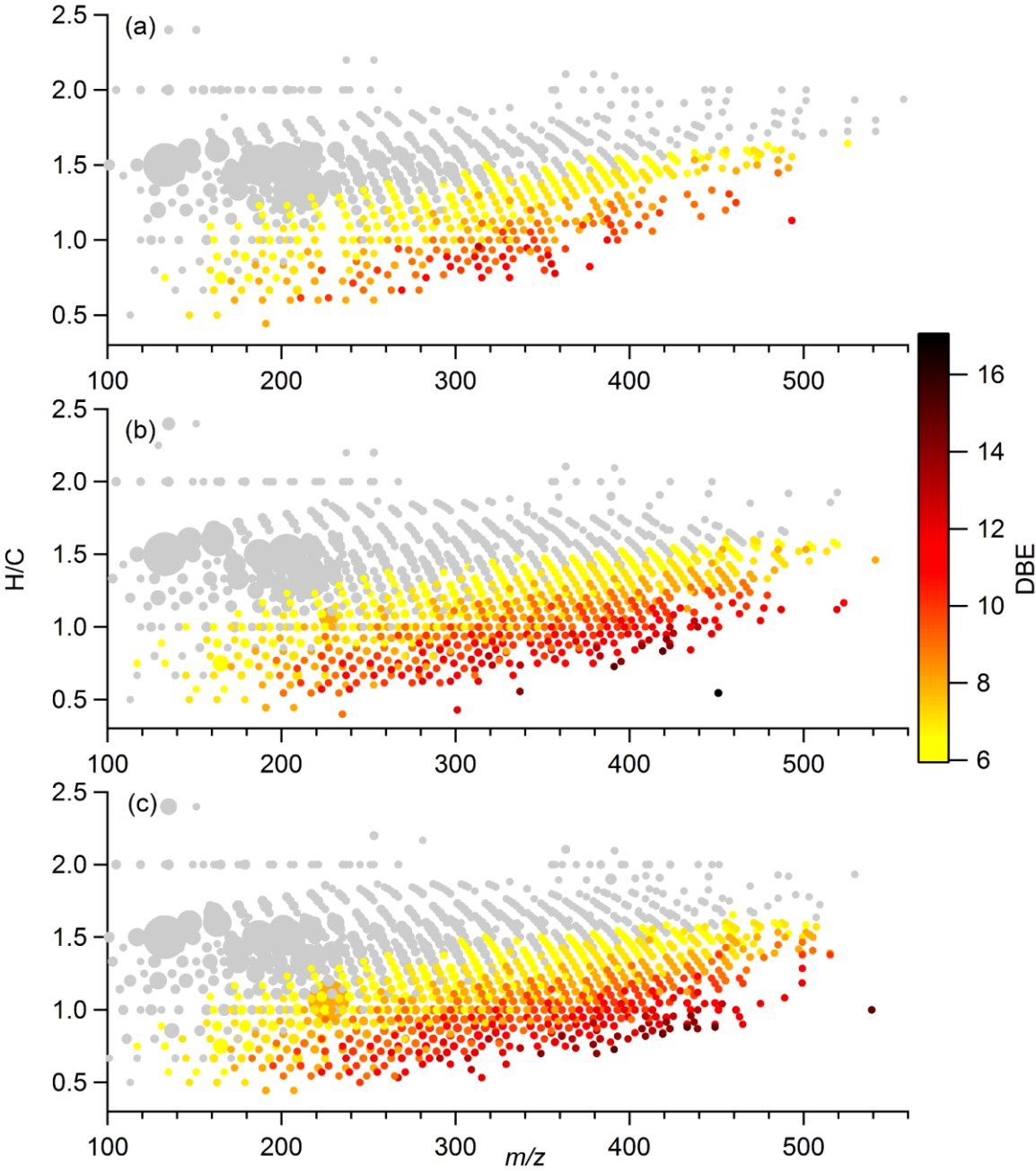

Figure 5. H/C vs m/z plot for CHO containing formulae in the samples from the periods with (a) low (b) moderately high and (c) very high incidents of fires. The marker areas reflect relative ion abundance in the sample. The colour code shows double bond equivalent (DBE) in the individual molecular formula. Molecular formulae with DBE<6 are shown as grey markers. The largest grey circles correspond to the ions at *m/z* 133.01425 (with neutral molecular formula $C_4H_6O_5$), *m/z* 187.0612 ($C_8H_{12}O_5$), *m/z* 201.07685 ($C_9H_{14}O_5$), *m/z* 203.05611 ($C_8H_{12}O_6$), and *m/z* 215.05611 ($C_9H_{12}O_6$).

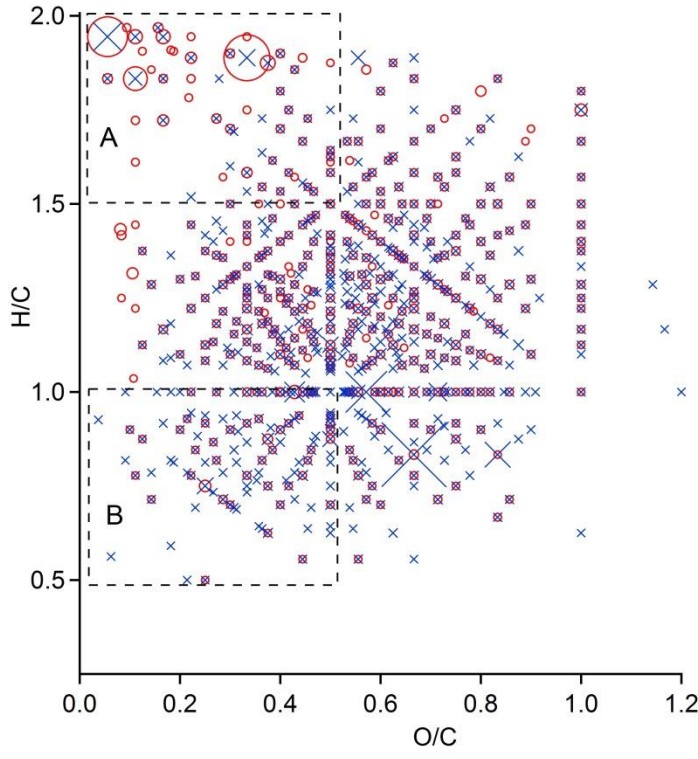


Figure 6. Overlaid Van Krevelen diagrams for CHON containing formulae in the samples
from the periods with low (red markers) and very high incidents (blue markers) of fires. The
marker areas reflect relative ion abundance in the sample. Areas 'A' and 'B' indicate
differences in the number of ions tentatively attributed to aliphatic and aromatic species,
respectively.

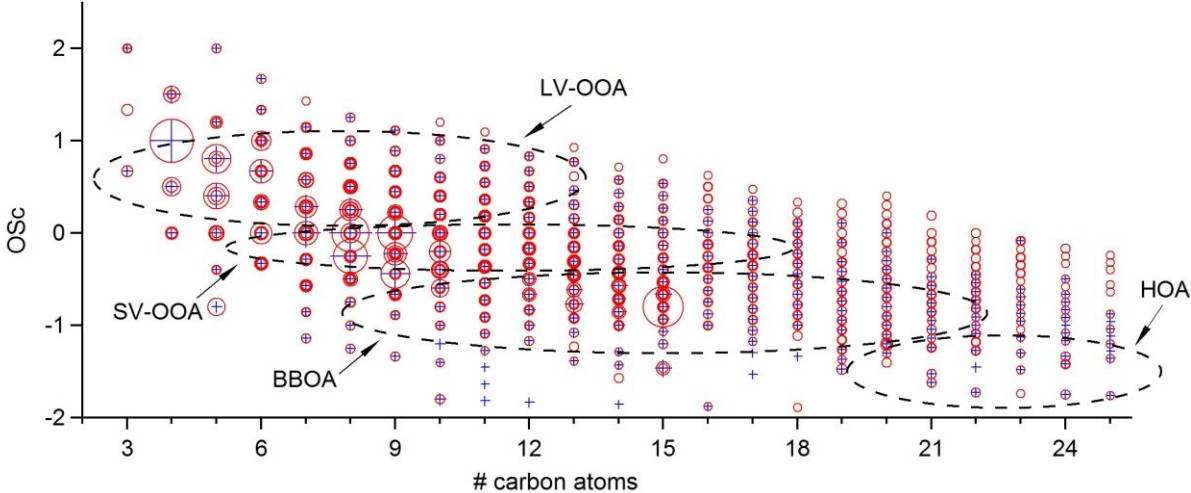

Figure 7. Overlaid carbon oxidation state (OSc) plots for CHO subgroups in the samples
from the periods with low (blue markers) and very high (red markers) incidents of fires. The
marker areas reflect relative ion abundance in the sample. The area marked as SV-OOA,
LV-OOA, BBOA and HOA correspond to the molecules associated with semivolatile and low-
volatility oxidised organic aerosol, biomass burning organic aerosol and hydrocarbon-like
organic aerosol as outlined by Kroll et al. (2011).

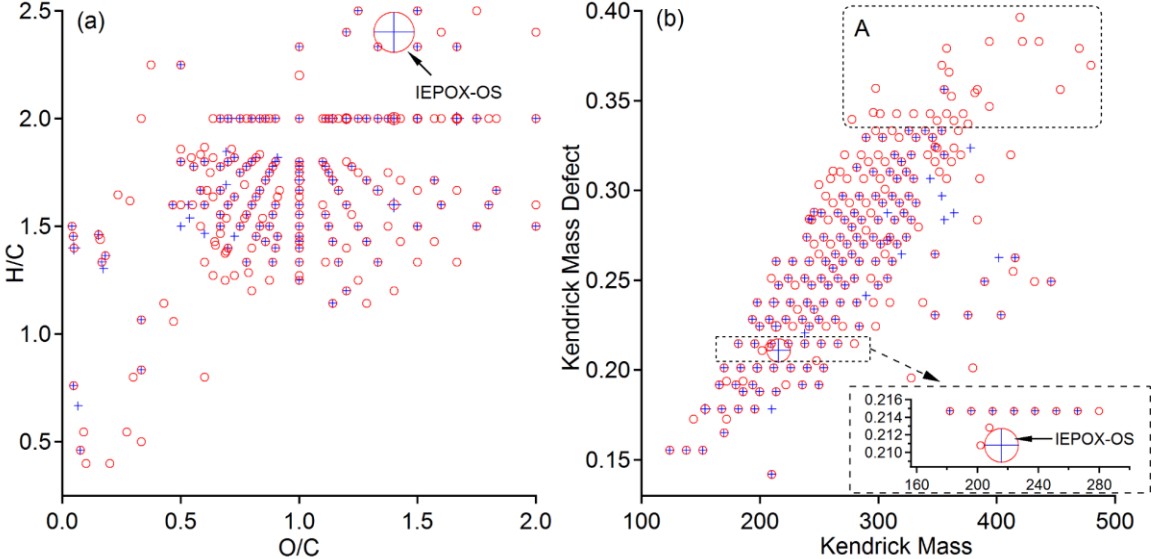


Figure 8. Overlaid Van Krevelen diagram (a) and Kendrick Mass Defect plot (b) for CHOS
containing formulae in the samples from the periods with low (blue markers) and very high
incidents of fires (red markers). The marker areas reflect relative ion abundance in the
sample. Red markers correspond to the ions from the period with the lowest incidents of
fires. Note that IEPOX-OS is not a part of any homologous series in the sample with very low
incident of fires and only one additional homologue in the sample that experienced very high
incident of fires (see enlarged area of the Fig 8a). Area 'A' in Kendrick Mass Defect (KMD)
plot shows formulae with KMD>0.33 that are mainly present in the sample with high incident
of fires.