# Peer review of "Molecular composition of organic aerosols in central Amazonia: an ultra-high resolution mass spectrometry study"

_Atmospheric Chemistry and Physics, 2016_

## Referee Comment (RC1) · Anonymous Referee #1 · 7 Jun 2016

General comments

The manuscript presents UHRMS data of PM2.5 aerosol samples collected in central Amazonia during both the 'wet' and 'dry' seasons. Several tracer compounds corresponding to sources of biogenic and anthropogenic organic aerosol (OA) have been tentatively identified and UHRMS visualisation tools such as the Kendrick mass defect, Van Krevelen diagrams etc have been plotted to obtain further information of the differences in molecular composition between samples. The work presented is very interesting, in particular the demonstrated change in the OA chemical composition with increasing number of incident fires. As the authors note, higher time resolution filter collection would have led to a better understanding of the various factors affecting

aerosol sources and formation at their sampling site. However, I do believe that this work offers new and interesting data. The manuscript did however suffer from inaccuracies with the incorrect use of acronyms, spelling mistakes and wording, which is in some places, is rather difficult to understand. I recommend that this manuscript should be published but only after the manuscript has been thoroughly checked for errors and the comments below, particularly specific comments 1 and 2 which are imperative to the work, have been addressed.

Specific comments

1. Lines 259 - 272. It appears that the identification of all of the tracer compounds with the exception of IEPOX (which is mentioned in the experimental section) was confirmed using only UHRMS via MS2. As stated, UHRMS does not differentiate between structural isomers. The authors should be able to confirm if the tracer compounds are present in the samples through the comparison of the ion fragmentation patterns to the literature or authentic standards of the tracer compounds, providing the fragmentation patterns are not too 'messy' (i.e. multiple fragmentations of different structural isomers). I suspect the fragmentation data of m/z 161.0456 consisting of four tracer compounds and possibly other structural isomers may be particularly difficult to interpret and this should be mentioned in the manuscript. In addition, I don't believe the authors can attribute the entire ion abundance of m/z 203.05611 to 3-MBTCA, unless the fragmentation data shows no indication of any other possible structural isomers. Finally, how do the authors know that other possible structural isomers are not largely contributing to the ion abundance of the other tracer compounds (i.e. $C_6H_5NO_4$ etc)? The authors need to provide more justification/evidence for the identification of these compounds and the use of their ion abundances in Figure 4.

2. Were the IOP1 and IOP2 samples analysed on the same day, or was the detector variation of the UHRMS monitored during the analysis period? The UHRMS will vary in sensitivity. Running samples days or weeks apart may result in a variation in the amount of species observed due to fluctuations in the UHRMS sensitivity (e.g. as the

mass spectrometer becomes 'dirty', the detector sensitivity will decrease, affecting the ion intensity and subsequently the amount of species observed). This is particularly important in Figures 2 and 4 where the molecular formulae and ion abundances, respectively, are compared. If the samples were not analysed at the same time or if the detector sensitivity was not monitored, the authors would not be able to compare the ion abundance of the tracer compounds as shown in Figure 4. This is also likely to affect the comparison of the molecular formulae in Figure 2. The work presented here would then be only qualitative (rather than semi-quantitative). Were any attempts made to account for variations in detector sensitivity?

3. Line 217 states that the number of molecular formulae of species containing CHO increased by $\sim$ 20% from IOP1 to IOP2, but Figure 2 shows that this increase is within the standard deviation of the three replicate measurements. Please can the authors state in line 217 that this $\sim$ 20% difference is based on the average number of molecular formulae. Can the authors demonstrate that these differences are statistically significant? Do the ratios of the compounds classes differ between wet and dry season?

4. The experimental section needs to be separated into sections to make it clearer. Currently, the direct infusion flow rate follows the LC-MS parameters after UHRMS has already been discussed (line 140). Sub-headings such as 'LC-MS analysis', 'ESI-UHRMS analysis' and 'data processing' would make the experimental section easier to understand.

5. Line 157, competitive ionisation is not the only reason why ion intensities do not reflect the concentration of the compounds when using ESI. The ionisation efficiency of species will also vastly differ depending largely on their chemical structure and composition (see Oss et al (2010)). Please can the authors acknowledge this in the manuscript?

6. The authors use very strict molecular formulae constraints, along with other parameters such as O/C ratio $\geq$ 1.3, 0.3 $\leq$ H:C ratio etc. This is likely to remove a large

proportion of the observed peaks from further analysis. I understand why the authors have done this, but please can they include the percentage of the observed mass spectral peaks which are assigned molecular formulae in the manuscript (i.e. 57 % of the observed mass spectral peaks were assigned a molecular formulae using the constraints; as shown in Woznaik et al 2008)?

7. The authors refer to Kourtchev et al (2013) and (2015) for further details regarding the processing of the UHRMS data. From these papers, it appears that the background ions are subtracted from samples. If so, please can the authors include this in this manuscript? This is an important part of the data processing and needs to be mentioned in this manuscript too.

8. Line 228-229. The authors state that the daytime %RH during IOP1 was 89%. This seems a little high based on the data shown in Figure SI2. Please state in the manuscript whether this is the average %RH or maximum. Also, have the authors calculated the %RH only during the filter sampling time periods? Given that the authors are justifying why there is increased number of organonitrates in the IOP1 samples the %RH should only refer to the filter sampling time periods.

9. Line 251 states that wet deposition of aged or processed aerosol cannot be only reason for the observed differences in OSc. If the aerosol had wet deposited, how would the authors of sampled this?

10. Can the authors add the isoprene gas-phase measurements into Figure 4? Use of replicate figures with 'a' (benzene overlaid) and 'b' (isoprene overlaid) may prevent the data from looking too busy.

11. Line 443. Can the authors give more justification as to why they think the observation of these highly oxygenated species are likely to be associated with molecules produced through homogenous photochemical ageing reactions? Compounds with $\sim$ 10 oxygen atoms are likely to be of relatively low volatility residing mainly in the particulate phase. Heterogeneous reactions would seem likely here.

12. Can the authors show the data points from IOP1 and IOP2 in different colours/shapes in Figure SI3?

13. Can the authors draw the categorises/sources of aerosol (i.e. SV-OOA, BBOA etc) onto Figure 3 as shown in Kroll et al (2011). This will make the data much easier to visualise when describing in the results section.

Reference - Oss et al., (2010) Anal. Chem. 82. 2865-2872

Technical corrections

1. OH should be written as OH or 'OH radical'

2. Line 67, the use of 'participate in heterogeneous chemical reactions in the atmosphere' doesn't make an awful lot of sense in this sentence, re-word or remove.

3. Line 70, for the most part, precursor and oxidant types will determine the composition of SOA formed, which will in turn determine the light absorbing properties of the SOA. Remove 'precursor and oxidant types' from this sentence or re-word.

4. NOx should be written as $NO_x$ (use of subscript)

5. Line 75, remove 'for example', this sentence does not follow the above.

6. Line 89, 'UHRMS have a mass resolution...' should be, 'UHRMS has a mass resolution....'

7. Line 97, need a comma after Shanghai.

8. Line 98, this sentence would read better as; 'UHRMS has proven to be extremely useful or a value tool/technique for assessing.....'

9. Line 104, Martin et al 2015 is not in the references, do you mean Martin et al 2016?

10. Line 105, the T3 site is 69.4 km from Manaus (Martin et al 2016), not 70 km. Change to $\sim$ 69 km or 69.4 km.

11. Line 113, could you make this a little clearer? '...passed over the single large city (Manaus)'

12. Supplementary material Table SI1, is there a reason why the time is reported as, for example, 7H47? If not, change column header to 'Time (UTC, HH:MM)' and remove 'H'.

13. Line 121, this sentence reads as if the sampling flow rate changes during sample collection. Re-word.

14. Line 123, how were the samples stored at -4 ËŽC?

15. Line 127, 'optima' is the name of the product not the grade, the grade is LC-MS. Change.

16. Line 128, how was the sample reduced to a volume of 200 $\mu$L, via a nitrogen line or evaporator? If the latter, please give details of manufacturer etc.

17. Line 154, define CID

18. Line 154, 'MSMS' should be written as MS/MS or MS2

19. Line 156, include the word 'time' in 'chromatographic elution' (i.e. chromatographic elution time or retention time)

20. Line 184, define E/N before abbreviating

21. Supplementary material SI1, explain what 'MP14-06' etc (displayed on the figures) refers to in the figure caption.

22. Line 209, states that the majority of ions were associated with molecules less than 500 Da but Figure 1 only goes up to m/z 500. Either show the full m/z scan range in Figure 1 or re-phase Line 209 (e.g. the majority of species were observed between m/z 100 to 400).

23. Line 212, 'fragile compounds'. Why are some compounds fragile? Please expand

or reference.

24. Line 213 is difficult to read. Re-word.

25. Figure 2; include 'IOP1' and 'IOP2' next to 'wet' and 'dry' season respectively in Figure 1 or the opposite in the figure caption.

26. Supplementary material Table SI1, make clear which samples are from wet and dry season.

27. Line 223, Table SI1, NOy should be written as $NO_y$ (use of subscript).

28. Line 227 and Figure SI2 caption, define 'RH'.

29. Figure SI2 caption, what is 'ARM'? Define. Should this be in the references?

30. Figure SI2, what are the dashed lines displaying? Explain in caption.

31. Line 228, use of 'IOP1' then 'wet season'. Please use either wet and dry or IOP1 and IOP2.

32. Line 230 and elsewhere, 'OSc' should be written as '$OS_c$'

33. Figure 3, please give a starting number of carbon atoms on the x-axis or start from zero.

34. Line 256, move reference to the end of the sentence.

35. Line 273, SOx should be written as $SO_x$ (use of subscript).

36. Line 390, change to 'a reduced number of ' or 'decreased number of'

37. Lines 400 and 401 are difficult to understand. Be more precise (e.g. ....difference in OSc is more pronounced with compounds containing more than 7 carbon atoms). 'Affected ions'? Re-word.

38. Line 375, change 'nitroartomatic' to 'nitro-aromatic'
39. Line 376, 'overplayed'?

Please see supplement for correct use of acronyms and abbreviations.

Please also note the supplement to this comment:
http://www.atmos-chem-phys-discuss.net/acp-2016-404/acp-2016-404-RC1-
supplement.pdf

---

## Referee Comment (RC2) · Anonymous Referee #2 · 13 Jun 2016

General comments

This manuscript deals with the molecular characterization of PM2.5 aerosol collected in Manaus, Brazil, which is impacted by regional biomass burning, mainly during the dry season, and anthropogenic pollution from the city. Advanced analytical ultra-high resolution MS-based tools are applied, which allow a comprehensive MS data evaluation and identification of molecular formulae. Several comments were already formulated in a first review, with which I could agree. This review will therefore be limited to additional comments. The manuscript contains indeed interesting and novel data on Amazonian fine aerosol, which could be elaborated and is worth publishing after suitable revision.

What I miss in the manuscript is a comparison with previous studies dealing with the

detailed molecular characterization of Amazonian fine aerosol.

A first example: the 2-methyltetrols have been measured in several studies (e.g., Claeys et al., ACP 10, 9319-9331, 2010); the highest levels were observed during the dry period which is characterized by biomass burning (and higher particle concentrations of sulfuric acid). This observation is in agreement with the results obtained in the present study, taking into account that 2-methyltetrol sulfates were converted to 2-methyltetrols during the GC/MS procedure with prior trimethylsilylation?

A second example: A study on Amazonian biomass burning aerosol (Claeys et al., Environ. Chem., 9, 273-284, 2012) using LC/MS led to the molecular characterization of several strongly UV-absorbing nitro-aromatic compounds, with 4-nitrocatechol and isomeric methyl-nitrocatechols being the most abundant ones. Nitrocatechols are mentioned in the current manuscript but no mention is made of the methyl-nitrocatechols, which are very important markers for biomass burning secondary organic aerosol (SOA), formed from m-cresol emitted during the fires. In addition, also several biogenic SOA markers were identified in the study of Claeys et al. (2012), including MBTCA, terebic acid, terpenylic acid, 2-hydroxyterpenylic acid, and azelaic acid.

Specific comments

Lines 129-131: Part of the samples was used for LC/MS analysis but no LC/MS results are presented in the current manuscript. It would be very relevant to provide LC/MS results and as such support molecular assignments. It would also be relevant to see whether the major compounds found in LC/MS correspond to the major ones with the semi-quantitative direct infusion approach used in the present work.

Figure 1: The base peak in panel (a) is at m/z 171 (terpenylic acid?), but this ion is not discussed in the manuscript. Has this ion been assigned? There are also other abundant ions of the CHO type in the region below m/z 200 which merit attention, such as m/z 157 (terebic acid?) and m/z 187 (2-hydroxyterpenylic acid, or azelaic acid?), and are likely biogenic SOA markers.

Lines 209-229: As already mentioned above, LC/MS results would be very useful to support the molecular assignments, more useful in my opinion than MS/MS data, which in the case of 2-methyltetrolsulfates provide limited structural information (only the bisulfate anion). Quite some emphasis is given to the number of molecular formulae containing CHO, CHON, CHOS, and CHONS. More emphasis could be given to the molecular characterization of the major species, taking into account that LC/MS analysis has been performed and reference can be made to the literature. This type of information will be of great interest to readers dealing with molecular characterization.

Lines 279-286: Here, the origin of benzene is discussed and it is argued that benzene has mainly an anthropogenic origin because it correlates well with CO. It is not very clear what is meant by "anthropogenic origin". Biomass burning for domestic purposes (e.g., cooking) in urban locations can also be regarded as an anthropogenic activity and this must be clarified in the manuscript. Benzene could very well have mainly a biomass burning origin. More detailed insights could be obtained by measuring other aromatic compounds, such as cresols, and acetonitrile, which are characteristic for biomass burning; a good correlation between benzene and cresols/acetonitrile would point to a biomass burning origin. A differentiation between an anthropogenic and a tropical biomass burning origin cannot easily be made and will remain problematic. See the following article and references cited therein: Iinuma et al., Environ. Sci. Technol. 2010, 44, 8453–8459.

Lines 371-373: It would be relevant to mention 4-nitrocatechol and isomeric methyl-nitrocatechols in the group of nitroaromatic compounds, since they are characteristic of biomass burning SOA; see Iinuma et al., Environ. Sci. Technol. 2010, 44, 8453–8459.

Lines 393-396: In addition, the CHON molecules identified by LC/MS in biomass burning OA from Amazonia showed O/C ratios below 0.7, i.e., 4-nitrocatechol ($C_6H_5O_4N$; O/C = 0.67), isomeric methyl-nitrocatechols ($C_7H_7O_4N$; O/C = 0.57), and isomeric dimethyl-nitrocatechols ($C_8H_9O_4N$; O/C = 0.50). Ref: Claeys et al., Environ. Chem. 9,

273-284, 2012.

Lines 436-437: Species with molecular formulae $C_5H_{10}O_7S$ (m/z 213) could also be due to organosulfates formed from the green leaf volatiles 2-E-pentenal, 2-E-hexenal, and 3-hexenal, and have recently been characterized as isomeric 3-sulfooxy-2-hydroxypentanoic acid and 2-sulfooxy-3-hydroxypentanoic acid. Ref. Shalamzari et al., ACP 16, 7135-7148, 2016. See also the corresponding discussion document, where the issue is raised that $C_5H_{10}O_7S$ species could be oxidation products of isoprene.

Figure 5: What do the large grey circles between m/z 120 – 240 represent? Please, explain in the legend of the figure and discuss in the main text.

Figure SI5: What do the large grey circles at around m/z 180 and 280 represent in panel (a)? What do the large grey circles between m/z 140 and 180 represent in panels (b) and (c)? Please, explain in the legend of the figure and discuss in the main text.

Figure SI6: It is evident from these figures that CHO compounds are present at significant abundances in the natural background. What species do the large grey circles represent in panels (a–c)?

Figure SI7: I wonder what the large yellow circles (panels (b) and (c)) between m/z 150 and 200 represent. Do they correspond to m/z 168 ($C_7H_6NO_4$) compounds, due to isomeric methyl-nitrocatechols, which are expected to be very prominent and most abundant in the samples from the dry biomass burning period? Looking at panel (a) I wonder what the large grey circles around m/z 190 and 380 represent? Please, explain in the legend of the figure and discuss it in the main text.

I found Figures SI5, SI6 and SI6 the more interesting figures in the manuscript, but unfortunately they ended up in the supplement. Please, consider to include them in the main text, perhaps leaving out some other figures and putting some emphasis on methyl-nitrocatechols, specific SOA markers for biomass burning. Other interesting (but less abundant) biomass burning SOA markers are m/z 182 ($C_8H_8NO_4$) compounds, corresponding to isomeric dimethyl-nitrocatechols.

Lines 473 – 477: Here, the authors indicate that future work is needed to better understand the quantitative contributions of the various factors to the aerosol composition at the T3 site and they suggest to analyze samples with higher sampling resolution. A better approach would be to also measure specific marker compounds more quantitatively by LC/MS or other methods, including biogenic SOA markers, and primary and secondary biomass burning markers, and apply a receptor modelling technique. See, for example, the recent study by de Oliveira Alves et al. (Atmos. Environ., 120, 277-285, 2015), where for a site in western Amazonia, i.e., Porto Velho, a distinction could be made between contributions from biomass burning, fossil fuel combustion and a mixed source to the PM10 mass.

---

## Author Comment (AC1) · 12 Aug 2016

We would like to thank Reviewer#1 for very helpful comments and suggestions. Point by point responses to these comments are listed in the supplement.

Please also note the supplement to this comment:
http://www.atmos-chem-phys-discuss.net/acp-2016-404/acp-2016-404-AC1-supplement.pdf

---

## Author Response (AR1)

We would like to thank Reviewer# 1 and #2 for very helpful comments and suggestions. All comments and suggestions have been considered. Point by point responses to these comments are listed below. The line numbers correspond to the final revised document (with accepted track changes) unless otherwise stated.

**Reviewer #1**

General comments The manuscript presents UHRMS data of PM2.5 aerosol samples collected in central Amazonia during both the 'wet' and 'dry' seasons. Several tracer compounds corresponding to sources of biogenic and anthropogenic organic aerosol (OA) have been tentatively identified and UHRMS visualisation tools such as the Kendrick mass defect, Van Krevelen diagrams etc have been plotted to obtain further information of the differences in molecular composition between samples. The work presented is very interesting, in particular the demonstrated change in the OA chemical composition with increasing number of incident fires. As the authors note, higher time resolution filter collection would have led to a better understanding of the various factors affecting aerosol sources and formation at their sampling site. However, I do believe that this work offers new and interesting data. The manuscript did however suffer from inaccuracies with the incorrect use of acronyms, spelling mistakes and wording, which is in some places, is rather difficult to understand. I recommend that this manuscript should be published but only after the manuscript has been thoroughly checked for errors and the comments below, particularly specific comments 1 and 2 which are imperative to the work, have been addressed.

Specific comments 1. Lines 259 - 272. It appears that the identification of all of the tracer compounds with the exception of IEPOX (which is mentioned in the experimental section) was confirmed using only UHRMS via MS2. As stated, UHRMS does not differentiate between structural isomers. The authors should be able to confirm if the tracer compounds are present in the samples through the comparison of the ion fragmentation patterns to the literature or authentic standards of the tracer compounds, providing the fragmentation patterns are not too 'messy' (i.e. multiple fragmentations of different structural isomers). I suspect the fragmentation data of m/z 161.0456 consisting of four tracer compounds and possibly other structural isomers may be particularly difficult to interpret and this should be mentioned in the manuscript. In addition, I don't believe the authors can attribute the entire ion abundance of m/z 203.05611 to 3-MBTCA, unless the fragmentation data shows no indication of any other possible structural isomers. Finally, how do the authors know that other possible structural isomers are not largely contributing to the ion abundance of the other tracer compounds (i.e. C6H5NO4 etc)? The authors need to provide more justification/evidence for the identification of these compounds and the use of their ion abundances in Figure 4.

We would like to mention that we were very cautious in interpreting direct infusion results. For instance, we initially stressed the fact that '*The structural or isomeric information is not directly obtained from the direct infusion analysis….*' (lines 259-260 in the original text). We also stated that '*It must be noted that due to competitive ionisation of analytes in the direct infusion ESI analysis of the samples with a very complex matrix (i.e., aerosol extracts), the ion intensities do not directly reflect the concentration of the molecules in the sample; therefore, data shown in this work is semi-quantitative*' (lines 157-160). *Moreover, we stated that 'Direct infusion analysis suffers from competitive ionisation in the complex matrices and thus comparing ion intensities across samples has*

*to be done with caution'* (lines 275-277). It must be noted that there is a large number of publications indicating and justifying the use of the direct infusion analysis for semi-quantitative purposes (see review by Nizkorodov et al., 2011). While the term "nitrophenols" does include all possible isomers, we agree that other compounds may contribute to a molecular formula assigned as 3-MBTCA. As confirmed by the LC/MS analysis of selected samples the compound assigned to $C_8H_{12}O_6$ molecular formula corresponds to MBTCA. However, in the revised version of the manuscript we emphasised again that molecular formula assigned as 3-MBTCA may also include other compounds: *'Moreover, other compounds with similar molecular composition present in the aerosol matrix may also contribute to the ion intensities of the discussed above compounds.'* We also replaced 'MBTCA' by a molecular formula $C_8H_{12}O_6$ in the Fig 4.

2. Were the IOP1 and IOP2 samples analysed on the same day, or was the detector variation of the UHRMS monitored during the analysis period? The UHRMS will vary in sensitivity. Running samples days or weeks apart may result in a variation in the amount of species observed due to fluctuations in the UHRMS sensitivity (e.g. as the mass spectrometer becomes 'dirty', the detector sensitivity will decrease, affecting the ion intensity and subsequently the amount of species observed). This is particularly important in Figures 2 and 4 where the molecular formulae and ion abundances, respectively, are compared. If the samples were not analysed at the same time or if the detector sensitivity was not monitored, the authors would not be able to compare the ion abundance of the tracer compounds as shown in Figure 4. This is also likely to affect the comparison of the molecular formulae in Figure 2. The work presented here would then be only qualitative (rather than semi-quantitative). Were any attempts made to account for variations in detector sensitivity?

The instrument was routinely calibrated before the analysis. It must be noted that in the current study we used a nanoESI source where each sample is processed using a separate ESI tip and nozzle, so there is no carryover between samples. All samples were analysed in a random order and within 48-hours after extraction (to minimise possible methylation; therefore, the observed differences could not be attributed to the instrument contamination.

3. Line 217 states that the number of molecular formulae of species containing CHO increased by ~ 20% from IOP1 to IOP2, but Figure 2 shows that this increase is within the standard deviation of the three replicate measurements. Please can the authors state in line 217 that this ~ 20% difference is based on the average number of molecular formulae. Can the authors demonstrate that these differences are statistically significant? Do the ratios of the compounds classes differ between wet and dry season?

The t-test demonstrated that there is a significant difference for individual subgroups (e.g., CHO) between two compared seasons (p=0.0092 and p=0.00007).

As requested the following statement has been added to the text: *'The Student's t-test showed that the observed difference for CHO (p=0.0092) and CHON (p=0.00007) subgroups between two seasons is statistically significant.'*

4. The experimental section needs to be separated into sections to make it clearer. Currently, the direct infusion flow rate follows the LC-MS parameters after UHRMS has already been discussed (line 140). Sub-headings such as 'LC-MS analysis', 'ESIUHRMS analysis' and 'data processing' would make the experimental section easier to understand.

As suggested the subheadings '*Direct infusion UHRMS analysis*' and '*LC-MS analysis*' have been added to the experimental section.

5. Line 157, competitive ionisation is not the only reason why ion intensities do not re- flect the concentration of the compounds when using ESI. The ionisation efficiency of species will also vastly differ depending largely on their chemical structure and composition (see Oss et al (2010)). Please can the authors acknowledge this in the manuscript?

This phenomenon, which is true for all mass spectrometry ionisation techniques, including hard ionisation techniques such as electron ionisation, is already covered by the matrix statement in the same sentence. As suggested by the reviewer, Oss et al reference has been added to the text.

6. The authors use very strict molecular formulae constraints, along with other parameters such as O/C ratio ≥ 1.3, 0.3 ≤ H:C ratio etc. This is likely to remove a large proportion of the observed peaks from further analysis. I understand why the authors have done this, but please can they include the percentage of the observed mass spectral peaks which are assigned molecular formulae in the manuscript (i.e. 57 % of the observed mass spectral peaks were assigned a molecular formulae using the constraints; as shown in Woznaik et al 2008)?

As suggested, we added a table SI2 (analogous to that in Wozniak et al., 2008) showing % occurrence of formula groups to all peaks assigned molecular formulae in the mass spectra during the two sampling periods.

7. The authors refer to Kourtchev et al (2013) and (2015) for further details regarding the processing of the UHRMS data. From these papers, it appears that the background ions are subtracted from samples. If so, please can the authors include this in this manuscript? This is an important part of the data processing and needs to be mentioned in this manuscript too.

As suggested, the following statement has been added to the text: '*The background spectra obtained from the procedural blanks were also processed using the rules mentioned above. The formulae lists of the background spectra were subtracted from those of the ambient sample and only formulae with a sample/blank peak intensity ratio ≥ 10 were retained*'. Lines 179-182 (see revised text)

8. Line 228-229. The authors state that the daytime %RH during IOP1 was 89%. This seems a little high based on the data shown in Figure SI2. Please state in the manuscript whether this is the average %RH or maximum. Also, have the authors calculated the %RH only during the filter sampling time periods? Given that the authors are justifying why there is increased number of organonitrates in the IOP1 samples the %RH should only refer to the filter sampling time periods.

Yes, the values correspond to the maximum and the minimum RH during the filter sampling periods. This has been now clarified in the text.

9. Line 251 states that wet deposition of aged or processed aerosol cannot be only reason for the observed differences in OSc. If the aerosol had wet deposited, how would the authors of sampled this?

Unfortunately, we do not fully understand this remark but we assume that this comment is a misunderstanding: we did not collect any precipitation or aerosol deposited due to wet deposition but only particles that were not scavenged by cloud or rain droplets.

10. Can the authors add the isoprene gas-phase measurements into Figure 4? Use of replicate figures with 'a' (benzene overlaid) and 'b' (isoprene overlaid) may prevent the data from looking too busy.

We intentionally did not show isoprene data in this manuscript because it will be published as an independent work.

11. Line 443. Can the authors give more justification as to why they think the observation of these highly oxygenated species are likely to be associated with molecules produced through homogenous photochemical ageing reactions? Compounds with ~ 10 oxygen atoms are likely to be of relatively low volatility residing mainly in the particulate phase. Heterogeneous reactions would seem likely here.

In this sentence we are referring to a literature study which also observed highly oxygenated species and suggested that they could be produced through homogenous photochemical ageing reactions. The exact formation mechanism for these species is still highly debatable as for most of them there are no chemical standards. We agree that heterogeneous reactions could possibly also lead to formation of such compounds. To clarify this, we removed the word 'homogeneous' from the statement.

12. Can the authors show the data points from IOP1 and IOP2 in different colours/shapes in Figure SI3?

As suggested different markers were used for IOP1 and IOP2 data points in Fig SI3.

13. Can the authors draw the categorises/sources of aerosol (i.e. SV-OOA, BBOA etc) onto Figure 3 as shown in Kroll et al (2011). This will make the data much easier to visualise when describing in the results section.

The main emphasis of the figure 3 is to show the shift in the carbon oxidation state from dry to wet seasons in organic aerosol *throughout the whole mass range*. Addition of the categories would make the plot very busy and difficult to visualise the shift in the OSc. However, as suggested by the reviewer, we added these categories to another carbon oxidation state plot in the revised Fig. 7 and updated the figure caption accordingly.

Reference - Oss et al., (2010) Anal. Chem. 82. 2865-2872

Technical corrections 1. OH should be written as OH or 'OH radical'

Corrected

2. Line 67, the use of 'participate in heterogeneous chemical reactions in the atmosphere' doesn't make an awful lot of sense in this sentence, re-word or remove.

The word 'heterogeneous' has been removed

3. Line 70, for the most part, precursor and oxidant types will determine the composition of SOA formed, which will in turn determine the light absorbing properties of the SOA. Remove 'precursor and oxidant types' from this sentence or re-word.

As suggested by the reviewer 'precursor and oxidant types' has been removed from this statement.

4. NOx should be written as NOx (use of subscript)

Corrected

5. Line 75, remove 'for example', this sentence does not follow the above.

We disagree with this comment, reaction between anthropogenic nitrogen oxides ($NO_x$) and sulfur dioxide ($SO_2$) with a range of BVOCs leading to formation of organic nitrates is an example of anthropogenic/biogenic interactions discussed in the above sentence.

6. Line 89, 'UHRMS have a mass resolution...' should be, 'UHRMS has a mass resolution....'

We are discussing several techniques here (e.g. Fourier transform ion cyclotron resonance MS and Orbitrap MS). To address the reviewer's comment, we replaced 'MS' with 'mass spectrometers'.

7. Line 97, need a comma after Shanghai.

Corrected

8. Line 98, this sentence would read better as; 'UHRMS has proven to be extremely useful or a value tool/technique for assessing.....'

As suggested, the sentence has been changed to '*UHRMS has proven to be extremely useful in assessing chemical properties of the SOA*'

9. Line 104, Martin et al 2015 is not in the references, do you mean Martin et al 2016?

Corrected

10. Line 105, the T3 site is 69.4 km from Manaus (Martin et al 2016), not 70 km. Change to ~ 69 km or 69.4 km.

Corrected

11. Line 113, could you make this a little clearer? '...passed over the single large city (Manaus)'

Corrected

12. Supplementary material Table SI1, is there a reason why the time is reported as, for example, 7H47? If not, change column header to 'Time (UTC, HH:MM)' and remove 'H'. 13.

Corrected

Line 121, this sentence reads as if the sampling flow rate changes during sample collection. Re-word.

The sentence has been change to '*The airflow through the sampler was approximately 10 L min$^{-1}$*'.

14. Line 123, how were the samples stored at -4 ĚŽC?

The sentence has been extended to *'…and stored in the freezer at −4°C until analysis.'*

15. Line 127, 'optima' is the name of the product not the grade, the grade is LC-MS. Change.

Optima is a trademark name of the Fisher LC/MS grade solvents which is different from the regular LC/MS grade solvents available on the market. The 'TM' and 'LC/MS' have been added to the revised version of the manuscript.

16. Line 128, how was the sample reduced to a volume of 200 µL, via a nitrogen line or evaporator? If the latter, please give details of manufacturer etc.

The sentence has been extended to *'..using a nitrogen line'*.

17. Line 154, define CID

Corrected

18. Line 154, 'MSMS' should be written as MS/MS or MS2

Corrected

19. Line 156, include the word 'time' in 'chromatographic elution' (i.e. chromatographic elution time or retention time)

Added

20. Line 184, define E/N before abbreviating

*'a field density ratio'* has been added before *'E/N'*

21. Supplementary material SI1, explain what 'MP14-06' etc (displayed on the figures) refers to in the figure caption.

As suggested, this has been now clarified in the figure caption: '*72 h back air mass history ('footprints') arriving at the T3 station for the periods of the analysed filters (labelled as e.g., MP14-06, MP14-16, MP14-17).*'Lines 37-39

22. Line 209, states that the majority of ions were associated with molecules less than 500 Da but Figure 1 only goes up to m/z 500. Either show the full m/z scan range in Figure 1 or re-phase Line 209 (e.g. the majority of species were observed between m/z 100 to 400).

The line 209 (in the original text) has been rephrased to '*……were associated with molecules below 500 Da although the measured mass goes up to 900 Da.*'

23. Line 212, 'fragile compounds'. Why are some compounds fragile? Please expand

The sentence has been extended to *'…(e.g. highly oxygenated compounds)'*.

24. Line 213 is difficult to read. Re-word.

As suggested the sentence has been rephrased to: '*The largest group of identified molecular formulae in all samples were attributed to molecules containing CHO atoms only (1051±141 formulae during IOP2 and 820±139 during IOP1), followed by CHON (537±71 during IOP2 and 329±71 during IOP1), CHOS (183±34 during IOP2 and 137±31 during IOP1) and CHONS (37±11 during IOP2 and 28±10 during IOP1) (Fig. 2).*'

25. Figure 2; include 'IOP1' and 'IOP2' next to 'wet' and 'dry' season respectively in Figure 1 or the opposite in the figure caption.

Corrected

26. Supplementary material Table SI1, make clear which samples are from wet and dry season.

This has been now clarified in the Table SI1 footnote*: The samples MP14_06 to MP14_28 correspond to 'wet' (IOP1) period and MP14_128 to MP14_153 to 'dry' (IOP2) period.*

27. Line 223, Table SI1, NOy should be written as NOy (use of subscript).

Corrected

28. Line 227 and Figure SI2 caption, define 'RH'.

Defined

29. Figure SI2 caption, what is 'ARM'? Define. Should this be in the references?

The ARM has been defined and the link to the website is provided: '*Figure SI2. Relative humidity (RH) at the T3 sampling site during (a) IOP1 and (b) IOP2 The arrows indicate sample collection periods. Atmospheric Radiation Measurement (ARM) data source http://www.archive.arm.gov.* '

30. Figure SI2, what are the dashed lines displaying? Explain in caption.

The explanation has been added to the figure legend: *'The continuous dashed line indicates the lowest and highest RH vales during both seasons'*

31. Line 228, use of 'IOP1' then 'wet season'. Please use either wet and dry or IOP1 and IOP2.

For consistency we have added IOP2 to this sentence: *'In this respect, while night time maximum RH during both filter sampling periods was very similar (~90%), day-time RH during IOP1 was higher (89%) compared to that from the IOP2 period (66%) (Fig. SI2).'*

32. Line 230 and elsewhere, 'OSc' should be written as 'OSc'

We do not understand this comment.

33. Figure 3, please give a starting number of carbon atoms on the x-axis or start from zero.

Done

34. Line 256, move reference to the end of the sentence.

Corrected

35. Line 273, SOx should be written as SOx (use of subscript).

Corrected

36. Line 390, change to 'a reduced number of ' or 'decreased number of'

As suggested we replaced 'reduced' by 'decreased'

37. Lines 400 and 401 are difficult to understand. Be more precise (e.g. ....difference in OSc is more pronounced with compounds containing more than 7 carbon atoms). 'Affected ions'? Re-word.

Yes, in this sentence we are discussing figure 7 and thus OSc differences associated with the carbon atoms in the molecular formulae. For clarity we expanded this sentence to *'The difference in OSc becomes even more pronounced with increased numbers of carbons (e.g. >7 carbon atoms) in the detected molecular formulae'.*

38. Line 375, change 'nitroartomatic' to 'nitro-aromatic'

Done

39. Line 376, 'overplayed'?

Changed to 'overlaid'

**Reviewer #2**

General comments: This manuscript deals with the molecular characterization of PM2.5 aerosol collected in Manaus, Brazil, which is impacted by regional biomass burning, mainly during the dry season, and anthropogenic pollution from the city. Advanced analytical ultra-high resolution MS-based tools are applied, which allow a comprehensive MS data evaluation and identification of molecular formulae. Several comments were already formulated in a first review, with which I could agree. This review will therefore be limited to additional comments. The manuscript contains indeed interesting and novel data on Amazonian fine aerosol, which could be elaborated and is worth publishing after suitable revision. What I miss in the manuscript is a comparison with previous studies dealing with the detailed molecular characterization of Amazonian fine aerosol.

A first example: the 2-methyltetrols have been measured in several studies (e.g., Claeys et al., ACP 10, 9319-9331, 2010); the highest levels were observed during the dry period which is characterized by biomass burning (and higher particle concentrations of sulfuric acid). This observation is in agreement with the results obtained in the present study, taking into account that 2-methyltetrol sulfates were converted to 2-methyltetrols during the GC/MS procedure with prior trimethylsilylation?

In the current study we concentrated on direct infusion mass spectrometry analysis and therefore we intentionally limited our comparison mainly to the literature that employed similar techniques. As suggested by the reviewer comparison to previous studies by Claeys et al. (2010) has been added to the text: *'This is also in agreement with previous studies from Amazon where the highest levels of 2-methyltetrols were observed during the dry period which was characterised by biomass burning*

*(and higher particle concentrations of sulfuric acid) (Claeys et al., 2010). Considering that Claeys et al (2010) employed alternative GC/MS procedure with prior trimethylsylilation, 2-methyltetrol sulfates were converted to 2-methyltetrols and not detectable as separate OS compounds.'* Lines 339-344 (see revised text)

A second example: A study on Amazonian biomass burning aerosol (Claeys et al., Environ. Chem., 9, 273-284, 2012) using LC/MS led to the molecular characterization of several strongly UV-absorbing nitro-aromatic compounds, with 4-nitrocatechol and isomeric methyl-nitrocatechols being the most abundant ones. Nitrocatechols are mentioned in the current manuscript but no mention is made of the methyl-nitrocatechols, which are very important markers for biomass burning secondary organic aerosol (SOA), formed from m-cresol emitted during the fires.

As suggested discussion of methyl-nitrocatechols has been added to the text: '*… and methyl-nitrocatechols ($C_7H_7NO_4$, m/z 168.03023 ) are important markers for biomass burning OA, formed from m-cresol emitted during biomass burning (Iinuma et al., 2010)*' Lines 283-285

In addition, also several biogenic SOA markers were identified in the study of Claeys et al. (2012), including MBTCA, terebic acid, terpenylic acid, 2-hydroxyterpenylic acid, and azelaic acid.

*Please see our response below.*

Specific comments Lines 129-131: Part of the samples was used for LC/MS analysis but no LC/MS results are presented in the current manuscript. It would be very relevant to provide LC/MS results and as such support molecular assignments. It would also be relevant to see whether the major compounds found in LC/MS correspond to the major ones with the semi-quantitative direct infusion approach used in the present work. Figure 1: The base peak in panel (a) is at m/z 171 (terpenylic acid?), but this ion is not discussed in the manuscript. Has this ion been assigned? There are also other abundant ions of the CHO type in the region below m/z 200 which merit attention, such as m/z 157 (terebic acid?) and m/z 187 (2-hydroxyterpenylic acid, or azelaic acid?), and are likely biogenic SOA markers.

To be able to compare mass spectral intensities from different aerosol samples, as well as to minimise matrix effect, we aimed to have similar aerosol loading of the sample extracts for the direct infusion analysis. Since we aimed for approximately 0.3 μg μL$^{-1}$ of particulate matter in each sample extract, there was enough aerosol material only for direct infusion analysis for most of the samples. Only a few aerosol samples had sufficient loading for both direct infusion and LC/MS analyses. Therefore, the later technique was only used to confirm the assignments of a few marker compounds observed in direct infusion analysis. This now has been clarified in the text:

*'Depending on the aerosol loading of the analysed samples a part (1/2 to whole) of the filter was extracted in methanol (Optima R LC/MS grade, Fisher Scientific) in a chilled ice slurry, filtered through a Teflon filter (0.2 μm, ISODiscTM Supelco) and reduced by volume using a nitrogen line to achieve approximately 0.3 μg of aerosol per μL methanol. Several samples with the highest aerosol loading were divided into two parts for both direct infusion and LC/MS analyses, while the samples with the lowest loading were only analysed using direct infusion analysis.'* Lines 126-132

With direct infusion analysis we identified more than a thousand molecular formulae, therefore the main emphasis was placed on a bulk molecular composition of the OA rather than identification of all possible marker compounds and which would be a study different to the one presented here. The discussion of the individual compounds was limited to a few known marker compounds that corresponded to the most intense ions.

The base peak in the panel (a) at $m/z$ 173.0454 corresponds to $C_7H_{10}O_5$, which is different from that of terpenylic acid. Unfortunately, this molecule was not clearly identified by the LC/MS analysis. With regards to the other abundant ions of the CHO type in the region below $m/z$ 200, an ion at $m/z$ 187.0612 corresponds to a deprotonated molecular formula $C_8H_{12}O_5$, which is neither 2-hydroxyterpenylic acid nor azelaic acid. An ion at $m/z$ 157.01425 corresponds to a deprotonated molecular formula $C_6H_6O_5$, which is also different from terebic acid. Unfortunately, due to the absence of standards for the above mentioned compounds, the discussion of these molecular formula would be highly speculative.

Lines 209-229: As already mentioned above, LC/MS results would be very useful to support the molecular assignments, more useful in my opinion than MS/MS data, which in the case of 2-methyltetrolsulfates provide limited structural information (only the bisulfate anion). Quite some emphasis is given to the number of molecular formulae containing CHO, CHON, CHOS, and CHONS. More emphasis could be given to the molecular characterization of the major species, taking into account that LC/MS analysis has been performed and reference can be made to the literature. This type of information will be of great interest to readers dealing with molecular characterization.

Please see our response above.

Lines 279-286: Here, the origin of benzene is discussed and it is argued that benzene has mainly an anthropogenic origin because it correlates well with CO. It is not very clear what is meant by "anthropogenic origin". Biomass burning for domestic purposes (e.g., cooking) in urban locations can also be regarded as an anthropogenic activity and this must be clarified in the manuscript. Benzene could very well have mainly a biomass burning origin. More detailed insights could be obtained by measuring other aromatic compounds, such as cresols, and acetonitrile, which are characteristic for biomass burning; a good correlation between benzene and cresols/acetonitrile would point to a biomass burning origin. A differentiation between an anthropogenic and a tropical biomass burning origin cannot easily be made and will remain problematic. See the following article and references cited therein: Iinuma et al., Environ. Sci. Technol. 2010, 44, 8453–8459.

We agree that the correlation between benzene and cresols/acetonitrile would provide more information on the aerosol sources; unfortunately, cresols/acetonitrile data is not available for our study. In Manaus for heating and cooking purposes people mainly use natural gas; therefore, a significant contribution from these activities to the biomass burning OA at the site is highly unlikely. This statement has been added to the text: *'In Manaus natural gas is mainly used for heating and cooking and therefore, the contribution from these activities to biomass burning OA at our site is highly unlikely'.* Lines 310-312

Lines 371-373: It would be relevant to mention 4-nitrocatechol and isomeric methylnitrocatechols in the group of nitroaromatic compounds, since they are characteristic of biomass burning SOA; see Iinuma et al., Environ. Sci. Technol. 2010, 44, 8453– 8459.

As suggested by the reviewer, the following statement has been added to the text: *'Nitro-aromatic compounds, such as nitrophenols (DBE=5) and N-heterocyclic compounds including 4-nitrocatechol and isomeric methyl-nitrocatechols are often observed in the OA from the biomass burning sources (Kitanovski et al., 2012a,b; Iinuma et al., 2010) and have been suggested as potential contributors to light absorption by brown carbon (Laskin et al., 2015).'* Lines 405-409

Lines 393-396: In addition, the CHON molecules identified by LC/MS in biomass burning OA from Amazonia showed O/C ratios below 0.7, i.e., 4-nitrocatechol (C6H5O4N; O/C = 0.67), isomeric methyl-nitrocatechols (C7H7O4N; O/C = 0.57), and isomeric dimethyl-nitrocatechols (C8H9O4N; O/C = 0.50). Ref: Claeys et al., Environ. Chem. 9273-284, 2012.

As suggested, the following statement has been added to the text: *'In addition, the CHON molecules identified by LC/MS in biomass burning OA from Amazonia showed O/C ratios below 0.7, i.e., 4-nitrocatechol ($C_6H_5NO_4$; O/C = 0.67), isomeric methyl-nitrocatechols ($C_7H_7NO_4$; O/C = 0.57), and isomeric dimethyl-nitrocatechols ($C_8H_9NO_4$; O/C = 0.50) (Claeys et al., 2012).'* Lines 434-437

Lines 436-437: Species with molecular formulae C5H10O7S (m/z 213) could also be due to organosulfates formed from the green leaf volatiles 2-E-pentenal, 2-Ehexenal, and 3-hexenal, and have recently been characterized as isomeric 3-sulfooxy2-hydroxypentanoic acid and 2-sulfooxy-3-hydroxypentanoic acid. Ref. Shalamzari et al., ACP 16, 7135-7148, 2016. See also the corresponding discussion document, where the issue is raised that C5H10O7S species could be oxidation products of isoprene.

The following statement has been added to the text: *'This molecular formula could also be associated with organosulfates (e.g., isomeric 3-sulfooxy-2-hydroxypentanoic acid and 2-sulfooxy-3-hydroxypentanoic acid) formed from the green leaf volatiles 2-E-pentenal, 2-E-hexenal, and 3-hexenal(Shalamzari et al., 2016)'* Lines 482-485

Figure 5: What do the large grey circles between m/z 120 – 240 represent? Please, explain in the legend of the figure and discuss in the main text.

As suggested by the reviewer the following explanation has been added to the text and the figure: *'The largest grey circles in Fig 5(a-c) correspond to the ions at m/z 133.01425 (with neutral molecular formula $C_4H_6O_5$), m/z 187.0612 ($C_8H_{12}O_5$), m/z 201.07685 ($C_9H_{14}O_5$), m/z 203.05611 ($C_8H_{12}O_6$), m/z 215.05611 ($C_9H_{12}O_6$) with DBE<6.'* Lines 367-370 and 881-883

FigureSI5: What do the large grey circles at around m/z180 and 280 represent in panel (a)? What do the large grey circles between m/z 140 and 180 represent in panels (b) and (c)? Please, explain in the legend of the figure and discuss in the main text.

As suggested by the reviewer the following explanation has been added to the text and the figure legend:

*'The largest grey circles in Figure SI5a correspond to the ions at m/z 187.11357 with a neutral molecular formula $C_9H_{17}NO_3$ and m/z 281.26459 with a neutral molecular formula $C_{18}H_{35}NO$. The largest grey circles in Figure SI5 b and c correspond to the ions at m/z 154.0146, m/z 168.03023 and m/z 152.03532 with neutral molecular formulae $C_6H_5NO_4$, $C_7H_7NO_4$ and $C_7H_7NO_3$, respectively'*. Lines 398-402 (main text) and 85-89 (SI).

Figure SI6: It is evident from these figures that CHO compounds are present at significant abundances in the natural background. What species do the large grey circles represent in panels (a–c)?

Figure SI6 shows exactly the same ion distribution as in Figure 5, but expressed using aromaticity index. The large circles in both figures correspond to the same molecular formula; therefore, to avoid repetition we added the explanation for these ions to only one of these figures.

Figure SI7: I wonder what the large yellow circles (panels (b) and (c)) between m/z 150 and 200 represent. Do they correspond to m/z 168 ($C_7H_6NO_4$) compounds, due to isomeric methyl-nitrocatechols, which are expected to be very prominent and most abundant in the samples from the dry biomass burning period? Looking at panel (a) I wonder what the large grey circles around m/z 190 and 380 represent? Please, explain in the legend of the figure and discuss it in the main text.

As suggested by the reviewer the following explanation has been added to the figure legend: '*The largest grey circles in panel 'a' correspond to ions at m/z 186.11357 and m/z 280.26459 with neutral molecular formulae $C_9H_{17}NO_3$ and $C_{18}H_{35}NO$, respectively. The yellow circles in panels 'b' and 'c' correspond to the ions at m/z 154.0146, m/z 168.03023 and m/z 152.03532 with molecular formulae $C_6H_5NO_4$, $C_7H_7NO_4$ and $C_7H_7NO_3$, respectively, which are known biomass burning marker compounds (see discussion in the main text)'.* Lines 105-110 (SI)

I found Figures SI5, SI6 and SI6 the more interesting figures in the manuscript, but unfortunately they ended up in the supplement. Please, consider to include them in the main text, perhaps leaving out some other figures and putting some emphasis on methyl-nitrocatechols, specific SOA markers for biomass burning. Other interesting (but less abundant) biomass burning SOA markers are m/z 182 ($C_8H_8NO_4$) compounds, corresponding to isomeric dimethyl-nitrocatechols.

A very large diversity of the data was produced in this work resulting in a large number of figures, so we had to be very selective which figures could be kept in the man text. Our justification was based on the fact that some of the figures in the SI would be rather challenging for a general reader with little mass spectrometry background. Therefore, we prefer to keep SI5 and SI6 in the SI.

As suggested by the reviewer the following explanation has been added to the text: '*It is worth mentioning that aerosol samples affected by biomass burning contained another interesting ion at m/z 182.04588 with a neutral molecular formula $C_8H_9NO_4$, possibly corresponding to biomass burning OA markers isomeric dimethyl-nitrocatechols (Kahnt et al., 2013)*'. Lines 409-412 (main text)

Lines 473 – 477: Here, the authors indicate that future work is needed to better understand the quantitative contributions of the various factors to the aerosol composition at the T3 site and they suggest to analyze samples with higher sampling resolution. A better approach would be to also measure specific marker compounds more quantitatively by LC/MS or other methods, including biogenic SOA markers, and primary and secondary biomass burning markers, and apply a receptor modelling technique. See, for example, the recent study by de Oliveira Alves et al. (Atmos. Environ., 120, 277285, 2015), where for a site in western Amazonia, i.e., Porto Velho, a distinction could be made between contributions from biomass burning, fossil fuel combustion and a mixed source to the PM10 mass.

[revised manuscript text omitted]

---

## Referee Report (RR1)

General comments

I recommend that this manuscript should be published in ACP after the few (minor) technical corrections listed below have been addressed.

Specific comments

1. Previous review, specific comment 2 - Were the IOP1 and IOP2 samples analysed on the same day, or was the detector variation of the UHRMS monitored during the analysis period? The UHRMS will vary in sensitivity. Running samples days or weeks apart may result in a variation in the amount of species observed due to fluctuations in the UHRMS sensitivity (e.g. as the mass spectrometer becomes 'dirty', the detector sensitivity will decrease, affecting the ion intensity and subsequently the amount of species observed). This is particularly important in Figures 2 and 4 where the molecular formulae and ion abundances, respectively, are compared. If the samples were not analysed at the same time or if the detector sensitivity was not monitored, the authors would not be able to compare the ion abundance of the tracer compounds as shown in Figure 4. This is also likely to affect the comparison of the molecular formulae in Figure 2. The work presented here would then be only qualitative (rather than semi-quantitative).Were any attempts made to account for variations in detector sensitivity?

Author response - The instrument was routinely calibrated before the analysis. It must be noted that in the current study we used a nanoESI source where each sample is processed using a separate ESI tip and nozzle, so there is no carryover between samples. All samples were analysed in a random order and within 48-hours after extraction (to minimise possible methylation; therefore, the observed differences could not be attributed to the instrument contamination.

Further review - The specific comment above was in regards to detector variation, not contamination or carryover, whilst both will also affect ion intensities. The authors have not fully addressed the question. It is still unclear if all the samples were analysed at the same time or if the detector variation was monitored. The authors note that the instrument is routinely calibrated, although I suspect (based on the method details) this only for mass accuracy and not for detector variation. The instrument should notify the user during mass calibration if the ion intensities of the calibrants are too low, highlighting sensitivity issues. However I do recommend in future, that the authors run samples at the same time when they plan to compare ion intensities (if not already done so) or use standards to monitor detection variation.

Technical corrections

1. Line 225 (previous review specific comment 23), why are some compounds fragile? Please expand or reference.
Author response - The sentence has been extended to '…(e.g. highly oxygenated compounds)'.
I don't agree that all highly oxygenated compounds are fragile. Please change to '(e.g. thermally labile)'.

2. Line 127, change the fraction '1/2' to the word 'half'.

3. Line 209, Change '10 000 particles' to '$10^4$ particles'

4. Line 261 and elsewhere; radical on OH not used throughout text. Please change all 'OH' to '˙OH'

5. Line 283, could methyl-nitrophenol $C_7H_7NO_3$ and methyl-nitrocatechol $C_7H_7NO_4$ also come from vehicular emissions (*i.e.* toluene oxidation products), given that the site is also affected by urban air pollution?

---

## Referee Report (RR2)

**Review ACPD: Kourtchev et al., version 3**

**Molecular composition of organic aerosols in central Amazonia: an ultra-high resolution mass spectrometry study**

The following technical corrections are suggested. It would also be best that the final manuscript is proofread by a native English speaker because there are still some problems with punctuation, such as, for example, the correct use of the comma (not listed in this report).

Line 52: ….. that of the wet period.

Line 53: …. from the forest …..

Line 62: ….., Andreae et al., 2015).

Line 77: …., nitrooxy-organosulfates …..

Line 83: …. (Nozière et al., 2015).

Line 89: ….. have a mass resolution ….

Line 92: ……, Nozière et al., 2015).

Line 157: LC-MS: make sure that "LC-MS" is used consistently throughout the text; sometimes use is made of "LC/MS".

Line 231: ….however, a rather insignificant …..

Line 236: …… during daytime and ……

Line 240: ….. consistent with recent studies, ….

Line 266: ….. during the wet …..

Line 270: …. from the OH-initiated oxidation ….

Line 272: …. from the ozonolysis reaction …..

Line 273: ….. of ion signal intensities (Note: ions are abundant; ion signals are intense; a field campaign could be intensive)

Line 293 and 295: ……ion signal intensities ….

Line 297: …. in the gas phase ….

Line 302: …. that the benzene …..

Line 314: ….., the number of forest …..

Lines 316, 320, and 323: …. ion signal intensities …..

Line 321: …. showed a very good …..

Line 324: ….. with the latter one ….

Lines 328, 333, and 334: …… ion signal intensity …..

Line 342: …. employed an alternative GC/MS …..

Line 349: ….., the highest CO ….

Line 361: ….. have a DBE value above ….

Line 396: …. with a pyrene core ….

Line 407: …. from biomass burning sources.

Line 411: …. corresponding to other biomass burning OA markers, i.e., isomeric dimethyl-nitrocatechols
….

Line 413: …. of nitro-aromatic compounds ….

Line 418: …. a significantly larger …..

Line 419: …. but a smaller number …..

Line 430: …. to result in a different ….

Line 440: …. towards a more oxidized state …

Line 447: …., the effect was much lower compared to that ….

Line 448: A higher number ….

Line 468: ….. in laboratory smog ….

Line 470: ….. in the presence of acidified …..

Line 475: …. are a useful …..

Line 479: …… most abundant ion at …..

Line 489: …. be associated with …..

Line 491: ….. that in most of the …..

Line 576: ….. in Rondônia, Brazil: ….

---

## Author Response (AR2)

Dear Editor,

All comments and suggestions have been considered. Point by point responses to these comments are listed below. The marked-up manuscript version is attached below.

**Referee report #1**

General comments

I recommend that this manuscript should be published in ACP after the few (minor) technical corrections listed below have been addressed.

Specific comments

1. Previous review, specific comment 2 - Were the IOP1 and IOP2 samples analysed on the same day, or was the detector variation of the UHRMS monitored during the analysis period? The UHRMS will vary in sensitivity. Running samples days or weeks apart may result in a variation in the amount of species observed due to fluctuations in the UHRMS sensitivity (e.g. as the mass spectrometer becomes 'dirty', the detector sensitivity will decrease, affecting the ion intensity and subsequently the amount of species observed). This is particularly important in Figures 2 and 4 where the molecular formulae and ion abundances, respectively, are compared. If the samples were not analysed at the same time or if the detector sensitivity was not monitored, the authors would not be able to compare the ion abundance of the tracer compounds as shown in Figure 4. This is also likely to affect the comparison of the molecular formulae in Figure 2. The work presented here would then be only qualitative (rather than semi-quantitative).Were any attempts made to account for variations in detector sensitivity?

Author response - The instrument was routinely calibrated before the analysis. It must be noted that in the current study we used a nanoESI source where each sample is processed using a separate ESI tip and nozzle, so there is no carryover between samples. All samples were analysed in a random order and within 48-hours after extraction (to minimise possible methylation; therefore, the observed differences could not be attributed to the instrument contamination.

Further review - The specific comment above was in regards to detector variation, not contamination or carryover, whilst both will also affect ion intensities. The authors have not fully addressed the question. It is still unclear if all the samples were analysed at the same time or if the detector variation was monitored. The authors note that the instrument is routinely calibrated, although I suspect (based on the method details) this only for mass accuracy and not for detector variation. The instrument should notify the user during mass calibration if the ion intensities of the calibrants are too low, highlighting sensitivity issues. However I do recommend in future, that the authors run samples at the same time when they plan to compare ion intensities (if not already done so) or use standards to monitor detection variation.

We disagree with this comment. The image current from oscillating ion packages in Orbitrap is recorded by the instrument over time and thus changes in sensitivity is not an issue in Orbitrap compared to many other MS detectors. Furthermore, when the instrument (i.e. ion optics) will get contaminated over time, it will adjust the injection time, so the absolute amount of ions in the Orbitrap detector for a given scan will always be the same (so the AGC target, unless the instrument is hitting the max IT). We assume the reviewer is referring to instruments like ToF, where a multi channel plate or a similar detector may age over time. Having said that, the routine calibration and transmission checks were performed which also included monitoring ion signal intensity. Moreover, blank samples were run after every 5 real samples, which showed no loss in the ion signal intensity. Considering that we run the analysis of our samples in three different mass ranges, in at least three analytical and instrumental replicates it is not feasible to perform the complete analysis of a large number of samples at 'the same time' (we assume the reviewer meant on the same day?).

The following statement has been added to the text: '*The calibration (as described in Kourtchev et al., 2013) and an ion transmission checks that include monitoring of ion signal intensity were routinely performed.*'

Technical corrections

1. Line 225 (previous review specific comment 23), why are some compounds fragile? Please expand or reference.  Author response - The sentence has been extended to '…(e.g. highly oxygenated compounds)'. I don't agree that all highly oxygenated compounds are fragile. Please change to '(e.g. thermally labile)'.

corrected

2. Line 127, change the fraction '1/2' to the word 'half'.

corrected

3. Line 209, Change '10 000 particles' to '104 particles'

corrected

4. Line 261 and elsewhere; radical on OH not used throughout text. Please change all 'OH' to '˙OH'

Word 'radical' has been added to OH throughout the text

5. Line 283, could methyl-nitrophenol C7H7NO3 and methyl-nitrocatechol C7H7NO4 also come from vehicular emissions (i.e. toluene oxidation products), given that the site is also affected by urban air pollution?

The vehicular source of methyl-nitrophenol C7H7NO3 and methyl-nitrocatechol C7H7NO4 has been added to the text: '*….as well as diesel exhaust.*'

**Reviewer #2**

Molecular composition of organic aerosols in central Amazonia: an ultra-high resolution mass spectrometry study

The following technical corrections are suggested. It would also be best that the final manuscript is proofread by a native English speaker because there are still some problems with punctuation, such as, for example, the correct use of the comma (not listed in this report).

The manuscript has been initially proofread by several native speakers (who are coauthors of this paper).

Line 52: ….. that of the wet period.

corrected

Line 53: …. from the forest …..

corrected

Line 62: ….., Andreae et al., 2015).

corrected

Line 77: …., nitrooxy-organosulfates …..

Do not see the difference with the initial statement

Line 83: …. (Nozière et al., 2015).

corrected

Line 89: ….. have a mass resolution ….

corrected

Line 92: ……, Nozière et al., 2015).

corrected

Line 157: LC-MS: make sure that "LC-MS" is used consistently throughout the text; sometimes use is made of "LC/MS".

corected

Line 231: ….however, a rather insignificant …..

corrected

Line 236: …… during daytime and ……

corrected

Line 240: ….. consistent with recent studies, ….

corrected

Line 266: ….. during the wet …..

Line 270: …. from the OH-initiated oxidation ….

corrected

Line 272: …. from the ozonolysis reaction …..

corrected

Line 273: ….. of ion signal intensities (Note: ions are abundant; ion signals are intense; a field campaign could be intensive)

corrected (the word 'signal' is often omitted in this context in mass spectrometry literature as it is trivial in this field).

Line 293 and 295: ……ion signal intensities ….

corrected

Line 297: …. in the gas phase ….

corrected

Line 302: …. that the benzene …..

We disagree with this suggestion

Line 314: ….., the number of forest ….

corrected

Lines 316, 320, and 323: …. ion signal intensities …..

corrected

Line 321: …. showed a very good …..

corrected

Line 324: ….. with the latter one ….

corrected

Lines 328, 333, and 334: …… ion signal intensity …..

corrected

Line 342: …. employed an alternative GC/MS …..

corrected

Line 349: ….., the highest CO ….

corrected

Line 361: ….. have a DBE value above ….

corrected

Line 396: …. with a pyrene core ….

corrected

Line 407: …. from biomass burning sources.

corrected

Line 411: …. corresponding to other biomass burning OA markers, i.e., isomeric dimethyl-nitrocatechols ….

corrected

Line 413: …. of nitro-aromatic compounds ….

corrected

Line 418: …. a significantly larger …..

corrected

Line 419: …. but a smaller number …..

corrected

Line 430: …. to result in a different ….

corrected

Line 440: …. towards a more oxidized state …

corrected

Line 447: …., the effect was much lower compared to that ….

corrected

Line 448: A higher number ….

corrected

Line 468: ….. in laboratory smog ….

Disagree, we are discussing the *specific* laboratory experiments published by Riva and co-authors.

Line 470: ….. in the presence of acidified …..

corrected

Line 475: …. are a useful …..

corrected

Line 479: …… most abundant ion at …..

corrected

Line 489: …. be associated with …..

corrected

Line 491: ….. that in most of the …..

corrected

Line 576: ….. in Rondônia, Brazil: ….

corrected

[revised manuscript text omitted]